# Perturbation Bounds for Low-Rank Inverse Approximations under Noise

**Phuc Tran**
VinUniversity

**Nisheeth K. Vishnoi**[*]
Yale University

## Abstract

Low-rank pseudoinverses are widely used to approximate matrix inverses in scalable machine learning, optimization, and scientific computing. However, real-world matrices are often observed with noise, arising from sampling, sketching, and quantization. The spectral-norm robustness of low-rank inverse approximations remains poorly understood. We systematically study the spectral-norm error $\|(\tilde{A}^{-1})_p - A_p^{-1}\|$ for an $n \times n$ symmetric matrix $A$, where $A_p^{-1}$ denotes the best rank-$p$ approximation of $A^{-1}$, and $\tilde{A} = A + E$ is a noisy observation. Under mild assumptions on the noise, we derive sharp non-asymptotic perturbation bounds that reveal how the error scales with the eigengap, spectral decay, and noise alignment with low-curvature directions of $A$. Our analysis introduces a novel application of contour integral techniques to the *non-entire* function $f(z) = 1/z$, yielding bounds that improve over naive adaptations of classical full-inverse bounds by up to a factor of $\sqrt{n}$. Empirically, our bounds closely track the true perturbation error across a variety of real-world and synthetic matrices, while estimates based on classical results tend to significantly overpredict. These findings offer practical, spectrum-aware guarantees for low-rank inverse approximations in noisy computational environments.

## 1 Introduction

Low-rank matrix approximations are foundational tools in machine learning, optimization, and scientific computing. They enable scalable algorithms by reducing memory and computation while preserving the dominant structure in high-dimensional data [24, 57]. A recurring task is to apply or approximate the inverse of a large symmetric (often positive semidefinite) matrix $A \in \mathbb{R}^{n \times n}$. Such inverse computations arise in kernel methods [19, 63], Gaussian processes [44], covariance-based inference [21, 33], and solvers for structured systems and graph Laplacians.

For large $n$, computing or storing the full inverse $A^{-1}$ is often infeasible. A common alternative is to approximate $A^{-1}$ with a low-rank surrogate. Let $A = \sum_{i=1}^{n} \lambda_i u_i u_i^\top$ be the eigendecomposition of $A$, with eigenvalues $\lambda_1 \geq \cdots \geq \lambda_n > 0$ and orthonormal eigenvectors $u_i \in \mathbb{R}^n$. Then $A^{-1} = \sum_{i=1}^{n} \lambda_i^{-1} u_i u_i^\top$, and the best rank-$p$ approximation of $A^{-1}$ in spectral norm is given by

$$A_p^{-1} := \arg\min_{\text{rank}(X) \leq p} \|A^{-1} - X\|_2 = \sum_{i=n-p+1}^{n} \lambda_i^{-1} u_i u_i^\top.$$

These directions capture the *flattest* (low-curvature) subspace, which dominates the condition number and affects stability in inverse-based algorithms. Such approximations are integral to fast solvers, adaptive preconditioners, and randomized linear algebra [13, 18], as well as optimization [43].

In many applications, however, one does not observe $A$ exactly but rather a noisy version $\tilde{A} = A + E$, where $E$ may arise from sampling error, sketching, quantization [3, 23], or deliberate perturbation

---

[*] Alphabetical order. Correspondence to `nisheeth.vishnoi@gmail.com`.

39th Conference on Neural Information Processing Systems (NeurIPS 2025).

for differential privacy [1, 61]. This naturally raises a basic question: *How robust is the rank-$p$ pseudoinverse to noise—how large is $\|(\tilde{A}^{-1})_p - A_p^{-1}\|$?*

This question is fundamental for assessing the stability of approximate solvers, Hessian-based preconditioners, and downstream learning or optimization pipelines that rely on noisy second-order information.

Classical matrix perturbation theory provides worst-case bounds for the full inverse:

$$\|\tilde{A}^{-1} - A^{-1}\| \leq \frac{\|A^{-1}\|^2 \|E\|}{1 - \|A^{-1}\|\|E\|}, \tag{1}$$

under the condition $\|A^{-1}\|\|E\| < 1$, or the first-order expansion

$$\tilde{A}^{-1} = A^{-1} - A^{-1}EA^{-1} + O(\|E\|^2);$$

see [26, 47]. Such perturbation bounds apply to arbitrary symmetric matrices $A$ and perturbations $E$, but they address the full inverse $A^{-1}$ and do not account for truncation to rank $p$. Moreover, they ignore the interaction between the noise and the eigengap, and scale poorly with $n$, often yielding overly pessimistic estimates. Structured identities such as the Sherman–Morrison–Woodbury formula apply only to specific low-rank updates, and recent results on low-rank perturbations under Schatten norms [35, 46, 57] do not provide spectral-norm guarantees for pseudoinverses under general noise.

**Our contributions.** We derive explicit, non-asymptotic spectral-norm perturbation bounds for the error $\|(\tilde{A}^{-1})_p - A_p^{-1}\|$, where $A_p^{-1}$ is the best rank-$p$ approximation of $A^{-1}$ and $\tilde{A} = A + E$ is a noisy observation. While certain bounds can be obtained from classical perturbation theory, our analysis yields *sharper, spectrum-adaptive guarantees* that depend explicitly on the eigengap $\delta_p := \lambda_{n-p} - \lambda_{n-p+1}$, the spectral decay of $A$, and the alignment of the perturbation $E$ with its low-curvature subspace. The main technical contribution is a novel application of *contour bootstrapping* to the non-entire function $f(z) = 1/z$, enabling localized resolvent expansions around the smallest eigenvalues and precise control over the perturbation of associated Riesz projectors [14, 27, 56].

Under the condition $4\|E\| \leq \min\{\lambda_n, \delta_{n-p}\}$, our main theorem establishes the bound

$$\|(\tilde{A}^{-1})_p - A_p^{-1}\| \leq 5\left(\frac{\|E\|}{\lambda_n^2} + \frac{\|E\|}{\lambda_{n-p}\,\delta_{n-p}}\right),$$

for positive-definite $A$ (Theorem 2.1); an extension to arbitrary real symmetric matrices appears in Section A. Our approach also applies to matrices with rank deficiency via their pseudoinverses (see Remark A.7). This result provides: (1) an explicit and interpretable spectral-norm guarantee; (2) up to a $\sqrt{n}$ improvement over classical inverse bounds in realistic regimes; and (3) a quantitative criterion for robustness of low-rank inverse approximations under general noise.

We empirically evaluate (i) the true perturbation error, (ii) our theoretical bound, and (iii) the estimate implied by classical Neumann-series and Eckart–Young–Mirsky analyses. Tests on both real and synthetic matrices—including sample covariance matrices, discretized elliptic operators, and sparse structural stiffness matrices (e.g., BCSSTK09)—show that our bound tracks the actual error within a small constant factor, whereas classical estimates often overpredict by one to two orders of magnitude (see Figures 1–2). These results provide a robust, interpretable certificate for the stability of noisy low-rank inverse approximations.

In Section 4.1, we evaluate the admissibility of the noise condition $4\|E\| < \min\{\lambda_n, \delta_{n-p}\}$ on standard datasets such as the 1990 US Census covariance and the BCSSTK09 stiffness matrices, showing that the resulting safety margins comfortably exceed noise levels common in differential-privacy and structural-engineering applications.

Section B presents an asymptotically refined bound for large-scale or synthetically structured matrices. Section E demonstrates a concrete application: we apply our framework to improve the theoretical convergence rate of the preconditioned conjugate-gradient (PCG) method. Using a low-rank–plus–regularization preconditioner, our bound yields tighter condition-number estimates and a provable $n^{1/4}$ improvement in the guaranteed iteration count compared with the classical Eckart–Young-Mirsky–Neumann-based analysis.

To the best of our knowledge, this is the first work to provide non-asymptotic spectral-norm bounds for $\|(\tilde{A}^{-1})_p - A_p^{-1}\|$ under general additive noise. Our analysis addresses a fundamental robustness

question in noisy inverse computation and complements existing algorithmic approaches by offering structural, spectrum-aware guarantees for when a low-rank inverse remains reliable.

**Related work.** Classical perturbation theory provides tools for analyzing the inverse of a perturbed matrix, but these focus on the full inverse and do not account for low-rank truncation or spectral structure [26, 47]. While the literature on low-rank approximation is extensive—spanning randomized SVD, projection methods, and sketching techniques [18, 24, 42, 57]—it primarily addresses approximation of $A$ itself or $f(A)$ when $f$ is a monotone-operator function[2], rather than inverse computations.

Several recent works study perturbations of low-rank approximations under Schatten or Frobenius norms [16, 35, 36, 37, 46], often in structured or sparse settings. However, these results do not apply to the spectral-norm error of low-rank inverse approximations. Similarly, while the Sherman-Morrison-Woodbury identity yields exact formulas for structured updates, it does not extend to arbitrary noise. Approximate Hessian inversion has been studied in the context of sketching [43], privacy [1, 61], and distributed optimization, but those works focus on convergence rather than the spectral stability of low-rank pseudoinverses.

Our technical approach builds on the contour integral representation of matrix functions, a classical tool in numerical analysis [27, 56]. While contour-based arguments have been used to analyze perturbations of spectral functionals associated with entire functions such as the matrix exponential (w.r.t. $f(z) = \exp(z)$) [39, 59] or the eigenspace-projection (w.r.t $f(z) = 1$) [31, 41, 53], they have rarely been applied to non-entire functions like the inverse. We adapt these techniques in a non-trivial way to $f(z) = 1/z$, localizing the resolvent expansion around small eigenvalues and bounding the impact of noise on the associated spectral projectors.

## 2   Theoretical results

For clarity, we present our main results in the case where $A$ is positive definite (PD). Extensions to general symmetric matrices are provided in Section A.

**Setup.** Let $A \in \mathbb{R}^{n \times n}$ be a real symmetric PD matrix with eigenvalues $\lambda_1 \geq \lambda_2 \geq \cdots \geq \lambda_n > 0$ and corresponding orthonormal eigenvectors $u_1, \ldots, u_n$. For each $1 \leq k \leq n-1$, define the eigenvalue gap

$$\delta_k := \lambda_k - \lambda_{k+1}.$$

Then $A^{-1}$ is also a real symmetric PD matrix, with eigenvalues $\lambda_n^{-1} \geq \lambda_{n-1}^{-1} \geq \cdots \geq \lambda_1^{-1} > 0$, and the same eigenvectors $u_n, \ldots, u_1$ (in reverse order).

Let $E \in \mathbb{R}^{n \times n}$ be a symmetric perturbation (error) matrix, and define the perturbed matrix as $\tilde{A} := A + E$. For a given rank $1 \leq p \leq n$, let $A_p^{-1}$ and $(\tilde{A}^{-1})_p$ denote the best rank-$p$ approximations of $A^{-1}$ and $\tilde{A}^{-1}$, respectively.

**Goal and classical baseline.** Our objective is to derive a spectral-norm bound on the difference between the best rank-$p$ approximations of $A^{-1}$ and $\tilde{A}^{-1}$:

$$\|(\tilde{A}^{-1})_p - A_p^{-1}\|.$$

While no prior results directly analyze this quantity, one can obtain a baseline estimate using classical tools: the Neumann expansion and the Eckart-Young-Mirsky (EYM) theorem [20]. Specifically, defining $E' := \tilde{A}^{-1} - A^{-1}$ and applying a low-rank approximation argument yields:

$$\|(\tilde{A}^{-1})_p - A_p^{-1}\| \leq 2(\|E'\| + \lambda_{n-p}^{-1}) \leq \frac{8\|E\|}{3\lambda_n^2} + \frac{2}{\lambda_{n-p}}, \tag{2}$$

valid when $4\|E\| \leq \lambda_n$; see Section D. This condition is needed for the application of Neumann expansion, and we refer to this bound as the *EYM–N bound*. This bound degrades when $\lambda_{n-p} \ll \lambda_n^2/\|E\|$ and fails to capture the limit $\|(\tilde{A}^{-1})_p - A_p^{-1}\| \to 0$ as $\|E\| \to 0$.

**Main result.** We now present a sharper, spectrum-adaptive bound based on contour bootstrapping. For clarity, we assume the eigenvalues of $A$ are ordered as: $+\infty = \lambda_0 > \lambda_1 \geq \lambda_2 \geq \cdots \geq \lambda_n > 0$.

---

[2] A matrix function $f$ is *monotone-operator* if, for all Hermitian $A, B$, $A - B$ is positive semi-definite implies $f(A) - f(B)$ is positive-semi definite. For example, $f(z) = z^t$ with $0 < t < 1$ is monotone-operator, whereas $f(z) = z^2$, $f(z) = e^z$, and $f(z) = 1/z$ are not.

**Theorem 2.1** (**Main perturbation bound for PD matrices**). *Let $A$ be a real symmetric PD matrix and $\tilde{A} = A + E$ with $E$ symmetric. If $4\|E\| \leq \min\{\lambda_n, \delta_{n-p}\}$, then*

$$\|(\tilde{A}^{-1})_p - A_p^{-1}\| \leq \frac{4\|E\|}{\lambda_n^2} + \frac{5\|E\|}{\lambda_{n-p}\delta_{n-p}}.$$

This bound consists of two interpretable components: the first term, $\|E\|/\lambda_n^2$, reflects classical perturbation scaling for the full inverse, while the second term, $\|E\|/(\lambda_{n-p}\delta_{n-p})$, captures the additional sensitivity introduced by projecting onto the subspace spanned by the smallest eigenvalues of $A$.

When the eigengap $\delta_{n-p} = \lambda_{n-p} - \lambda_{n-p+1}$ is well-separated and $\lambda_{n-p}$ is not too small, the low-rank approximation remains stable under noise, and the bound remains tight. Compared to classical bounds, this result explicitly accounts for spectral structure and subspace alignment, providing a more accurate estimate of the low-rank inverse perturbation.

Note that for $p = n$, we recover the full inverse case: $(\tilde{A}^{-1})_p = \tilde{A}^{-1}$ and $A_p^{-1} = A^{-1}$. In this setting, $\delta_{n-p} = \lambda_0 = +\infty$, so the second term vanishes and the bound simplifies to $\Theta(\|E\|/\lambda_n^2)$, recovering the Neumann bound.

**The gap condition.** The first assumption, $\|E\| \leq \lambda_n$, ensures that $\tilde{A}$ is invertible and $\tilde{A}^{-1}$ is well-defined. This matches the classical Neumann expansion, which fails when $\|A^{-1}\|\|E\| \geq 1$.

The second assumption, $4\|E\| \leq \delta_{n-p}$, which we call the *gap assumption*, ensures that the spectral ordering of eigenvalues is preserved under perturbation. By Weyl's inequality [62], this guarantees that the eigenvectors associated with the smallest $p$ eigenvalues of $\tilde{A}$ remain aligned with those of $A$, thereby preserving the low-rank inverse structure. When this assumption fails—i.e., when $\delta_{n-p} \ll \|E\|$—the eigenvalues of $\tilde{A}$ can reorder, leading to instability in the low-rank approximation; see Section I.1 for a concrete example.

**Our bound for a random matrix noise model.** If $E$ is a Wigner matrix (i.e., symmetric with i.i.d. sub-Gaussian entries), then $\|E\| = (2 + o(1))\sqrt{n}$ with high probability. Substituting this into Theorem 2.1 yields:

$$\|(\tilde{A}^{-1})_p - A_p^{-1}\| = O\left(\frac{\sqrt{n}}{\lambda_n^2} + \frac{\sqrt{n}}{\lambda_{n-p}\delta_{n-p}}\right).$$

In contrast, the EYM–N bound gives $O(\sqrt{n}/\lambda_n^2 + 1/\lambda_{n-p})$, which is larger when $\sqrt{n} \ll \delta_{n-p}$.

**Comparison to the EYM–N bound.** The EYM–N and bootstrapped bounds *coincide* in order of magnitude when $\|E\| \gg \lambda_n^2/\lambda_{n-p}$. In contrast, when $\|E\| \ll \lambda_n^2/\lambda_{n-p}$, our bound is smaller by a factor of

$$\min\left\{\frac{\lambda_n^2}{\lambda_{n-p}\|E\|}, \frac{\delta_{n-p}}{\|E\|}\right\}.$$

This "gain regime" arises naturally whenever either $p < \mathrm{sr}(A^{-1}) := \sum_{i=1}^n \lambda_n/\lambda_i$ or $\delta_{n-p}\lambda_{n-p} \ll \lambda_n^2$, i.e., $\min\{\lambda_n, \delta_{n-p}\} \ll \lambda_n^2/\lambda_{n-p}$, assuming the conditions of Theorem 2.1 are met. Notably, our bound becomes increasingly sharp as the noise level decreases.

In favorable cases, our result yields up to a $\sqrt{n}$-factor improvement. For example, consider a matrix $A$ with spectrum $\{n, 2n, \ldots, 10n, 20n, 20n, \ldots, 20n\}$ and $p = 10$. If $E$ is standard Gaussian noise, the EYM–N bound evaluates to $O\left(\frac{\sqrt{n}}{n^2} + \frac{1}{n}\right) = O(1/n)$, while our bound gives $O(\sqrt{n}/n^2) = O(n^{-3/2})$, demonstrating the expected $\sqrt{n}$-level gain. Section 4.2 empirically confirms that our bound consistently tracks the true error within a small constant (typically below 10), and outperforms the EYM–N estimate across both synthetic and real datasets.

**Applicability of assumptions.** Unlike the EYM–N bound, Theorem 2.1 additionally requires the gap condition $4\|E\| < \delta_{n-p}$. This assumption holds across a range of practically relevant matrix classes. For example, suppose $A = M^\top M$ is a sample covariance matrix, where $M \in \mathbb{R}^{m \times n}$ ($m \geq n$, ensuring that $A^{-1}$ is well-defined), and $E$ is a symmetric matrix with i.i.d. sub-Gaussian entries of mean zero and variance $\Delta^2$. If $\|M\|_F^2 \geq m \log n$ and $m > \frac{Cn^{3/2}\Delta}{\log n}$ for some constant $C > 0$, then both the spectral and gap conditions of Theorem 2.1 hold with high probability.

In Section 4.1, we compute $\lambda_n$ and $\delta_{n-p}$ for several real-world matrices $A$, and determine the maximum noise level $\|E\|$ for which the assumptions remain valid. Our findings show that both conditions—$4\|E\| \leq \lambda_n$ and $4\|E\| \leq \delta_{n-p}$—are satisfied robustly across many datasets.

In practice, exact verification of these assumptions is often unnecessary: as long as the estimation errors in $\lambda_n$ and $\delta_{n-p}$ are within $\|E\|$, our bound remains valid up to a constant factor (Step 3, Section 3). Thus, the assumptions are robust to moderate misestimation and allow scalable application in large-scale settings.

**Remark 2.2** (**A stronger but more technical bound**). *In the intermediate regime where $\lambda_n^2/\lambda_{n-} \ll \|E\| \ll \min\{\delta_{n-p}, \lambda_n\}$, the bound in Theorem B.2 (Section B) offers an asymptotic improvement over both the simple bound of Theorem 2.1 and EYM–N bound. This refinement is more technical and depends on additional structural quantities, such as the alignment of $E$ with the low-curvature eigenspace. While we do not empirically evaluate this bound, it may provide tighter guarantees in settings where noise is moderate and the spectral decay of $A^{-1}$ is slow.*

## 3    Proof overview

This section delineates the proof framework for Theorem 2.1, organized into three core stages. First, employing contour integration, we bound the perturbation by

$$\|(\tilde{A}^{-1})_p - A_p^{-1}\| \leq F := \tfrac{1}{2\pi\mathbf{i}}\|\int_\Gamma z^{-1}[(zI - \tilde{A})^{-1} - (zI - A)^{-1}]\|\,|dz|.$$

Here $\Gamma$ is a contour on the complex plane, encircling the $p$-bottom eigenvalues of $A$ and $\tilde{A}$. Unlike the Eckart–Young-Mirsky–Neumann (EYM–N) bound (see Section D), this formulation preserves the delicate $A - E$ interaction. Secondly, we develop the contour bootstrapping technique (Lemma 3.1), which under the assumption $4\|E\| \leq \min\{\lambda_n, \delta_{n-p}\}$, yields $F \leq 2F_1$ with

$$F_1 := \tfrac{1}{2\pi}\int_\Gamma \|z^{-1}(zI - A)^{-1}E(zI - A)^{-1}\|\,|dz|.$$

This bootstrapping argument, crafted specifically for the non-entire function $f(z) = 1/z$, replaces classical series expansions by a quantity that can be computed directly. Third, we construct a bespoke contour $\Gamma$— one specifically tailored so that the bottom-$p$ eigenvalues of $A$ and $\tilde{A}$ lie at prescribed distances from its sides. This tailored geometry renders the integral defining $F_1$ tractable and essentially tight, culminating in a sharp perturbation bound.

**Step 1: Representing the perturbation $\|(\tilde{A}^{-1})_p - A_p^{-1}\|$ via classical contour method.** Let $\lambda_1 \geq \cdots \geq \lambda_n > 0$ be the eigenvalues of $A$ with eigenvectors $u_i$. Then, $A^{-1}$ is well-defined, with eigenvalues $\lambda_n^{-1} \geq \lambda_{n-1}^{-1} \geq \cdots \geq \lambda_1^{-1} > 0$. Let $\tilde{\lambda}_1 \geq \cdots \geq \tilde{\lambda}_n > 0$ denote the eigenvalue of $\tilde{A}$. By Weyl's inequality [62],

$$\|E\| \ \geq \ |\lambda_n - \tilde{\lambda}_n| \ \geq \ \lambda_n - \tilde{\lambda}_n.$$

Under the assumption $4\|E\| \leq \lambda_n$ of Theorem 2.1, we obtain

$$\tilde{\lambda}_n \geq \lambda_n - \|E\| \geq 3\|E\| > 0.$$

Hence $\tilde{A}$ is also positive definite, and $\tilde{A}^{-1}$ is well-defined with eigenvalues $\tilde{\lambda}_n^{-1} \geq \tilde{\lambda}_{n-1}^{-1} \geq \cdots \geq \tilde{\lambda}_1^{-1} > 0$.

We now present the contour method to bound the perturbation of low-rank approximations of inverses in the spectral norm. Let $\Gamma$ be a contour in $\mathbb{C}$ that encloses $\lambda_n, \lambda_{n-1}, \ldots, \lambda_{n-p+1}$ and excludes $0$ and $\lambda_1, \lambda_2, \ldots, \lambda_{n-p}$. Thus, $f(z) = 1/z$ is analytic on the whole interior and boundary of $\Gamma$, and hence the contour integral representation [27, 30, 47] gives us:

$$\tfrac{1}{2\pi\mathbf{i}}\int_\Gamma z^{-1}(zI - A)^{-1}dz = \sum_{n-p+1 \leq i \leq n} \lambda_i^{-1}u_i u_i^\top = A_p^{-1}.$$

Here and later, $\mathbf{i}$ denotes $\sqrt{-1}$. The assumption $4\|E\| < \min\{\delta_{n-p}, \lambda_n\}$ and the construction of $\Gamma$ (see later this section) ensure that the eigenvalues $\tilde{\lambda}_i$ for $n \geq i \geq n - p + 1$ lie within $\Gamma$, while $0$ and all $\tilde{\lambda}_j$ for $j \leq n - p$ remain outside. We obtain the similar contour identity for $\tilde{A}$:

$$\tfrac{1}{2\pi\mathbf{i}}\int_\Gamma z^{-1}(zI - \tilde{A})^{-1}dz = \sum_{n \geq i \geq n-p+1} \tilde{\lambda}_i^{-1}\tilde{u}_i \tilde{u}_i^\top = (\tilde{A}^{-1})_p.$$

Thus, we obtain the *contour inequality*:

$$\|(\tilde{A}^{-1})_p - A_p^{-1}\| \leq F := \tfrac{1}{2\pi}\int_\Gamma \|z^{-1}[(zI - \tilde{A})^{-1} - (zI - A)^{-1}]\| \,|dz|.$$

This inequality makes the $A - E$ interaction explicit, but obtaining a sharp bound on its right-hand side remains a formidable analytical challenge.

**Step 2: Bounding $F \le 2F_1$ via contour bootstrapping method for non-entire function $f(z) = 1/z$.** By repeatedly applying the resolvent formula, one can expand

$$z^{-1}[(zI - \tilde{A})^{-1} - (zI - A)^{-1}] = \sum_{s=1}^{\infty} z^{-1}(zI - A)^{-1}[E(zI - A)^{-1}]^s.$$

This yields the bound:

$$F \le \sum_{s=1}^{\infty} F_s, \text{ where } F_s = \frac{1}{2\pi} \int_{\Gamma} \left\| z^{-1}(zI - A)^{-1}[E(zI - A)^{-1}]^s \right\| \, |dz|.$$

The traditional approach [30] attempted to estimate $F_s$ for each $s$. One can bound $F_s$ by

$$O\left( \|E\|^s \int_{\Gamma} \frac{|dz|}{|z| \min_{i \in [n]} |z - \lambda_i|^{s+1}} \right) = O\left[ \frac{\|E\|^s M_{\Gamma}}{\bar{\delta}^{s+2}} \right],$$

in which $\bar{\delta} := \min_{z \in \Gamma, i \in [n]} \{|z|, |z - \lambda_i|\}$ and $M_{\Gamma}$ is the total length of $\Gamma$. This traditional approach, with appropriate choices of $\Gamma$, can only provide a bound of $O\left( \|E\|/\lambda_n^2 + \|E\|/\delta_{n-p}^2 \right)$.

Moreover, when $f(z) = 1$ as in [53] or when $f$ is an entire function as in [52], the dominant contribution to $F$ arises from the term $F_1$, i.e., $F = O(F_1)$. We show that this relationship continues to hold for the rational case $f(z) = 1/z$ under the assumption $4\|E\| \le \min\{\lambda_n, \delta_{n-p}\}$.

**Lemma 3.1 (Contour Bootstrapping).** *If $4\|E\| \le \min\{\lambda_n, \delta_{n-p}\}$, then*

$$F \le 2F_1 = \frac{1}{\pi} \int_{\Gamma} \left| z^{-1}(zI - A)^{-1} E(zI - A)^{-1} \right| \, |dz|.$$

For entire functions $f$, the perturbation depends only on the top $p$ singular values of $A$, and the contour $\Gamma$ is chosen to isolate the leading eigenvalues $\{\lambda_1, \ldots, \lambda_p\}$. In contrast, the rational function $f(z) = 1/z$ requires the contour to enclose the smallest eigenvalues of $A$ while avoiding the singularity at $z = 0$.

This non-entire setting introduces two significant technical challenges. First, the relevant spectral components lie in the *smallest* $p$ eigenvalues of $A$, which are much more sensitive to perturbation. Indeed, when $\tilde{A}$ is a deformed Wigner matrix, $\|\tilde{A}^{-1}\| = O(n)$ with high probability for any fixed real $A$; see [29, 45, 48]. In such cases, the smallest singular values of $A$ are effectively destroyed by noise, illustrating the instability of low-curvature directions. Moreover, the perturbation of the low-rank approximation $\|\tilde{A}_p - A_p\|$ does not control the inverse approximations; see Section I.2 for a concrete counterexample. Second, constructing $\Gamma$ to isolate these low-lying eigenvalues while maintaining analyticity of $f(z) = 1/z$ requires additional care in bounding the associated resolvent terms.

**Step 3: Construction of $\Gamma$, $F_1$-estimation and proof completion of Theorem 2.1.** Now we show how Lemma 3.1, along with a careful choice of a contour $\Gamma$ can be used to prove Theorem 2.1. We need to construct the contour $\Gamma$ so that (i) the lowest $p$-eigenvalues of $A$ and $\tilde{A}$ lie inside and remain aligned, (ii) every point $z \in \Gamma$ is at least[3] $\delta_{n-p}/2$ or $\lambda_n/2$ from the spectrum of $A$, and (iii) the integral with respect to the resulting geometry is finite and computationally tractable. Indeed, the contour $\Gamma$ is set as a rectangle with vertices $(x_0, T), (x_1, T), (x_1, -T), (x_0, -T)$, where $x_0 := \lambda_n/2, x_1 := \lambda_{n-p+1} + \frac{\delta_{n-p}}{2}, T := 2\lambda_1$. Then, we split $\Gamma$ into four segments:

- Vertical segments: $\Gamma_1 := \{(x_0, t) | -T \le t \le T\}; \Gamma_3 := \{(x_1, t) | T \ge t \ge -T\}$.
- Horizontal segments: $\Gamma_2 := \{(x, T) | x_0 \le x \le x_1\} ; \Gamma_4 := \{(x, -T) | x_1 \ge x \ge x_0\}$.

Given the construction of $\Gamma$, we have

$$2\pi F_1 = \sum_{k=1}^{4} M_k, \text{ where } M_k := \int_{\Gamma_k} \left\| \sum_{n \ge i, j \ge 1} \frac{1}{z(z - \lambda_i)(z - \lambda_j)} u_i u_i^{\top} E u_j u_j^{\top} \right\| |dz|.$$

---

[3] The factor $1/2$ may be replaced by any fixed constant $c \in (0, 1)$ by adjusting the contour $\Gamma$, and the estimate changes only up to a constant. This flexibility makes the bound robust to moderate misestimation of $\lambda_n$ and $\delta_{n-p}$ in practice.

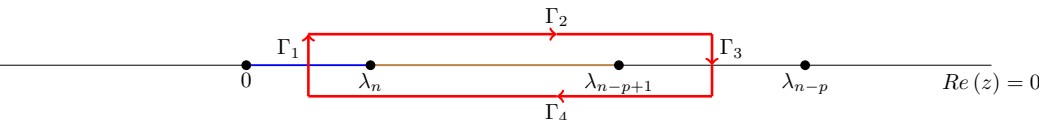

Note that the setting of the height $T = 2\lambda_1$ ensures that the integral does not blow up, and at the same time, the main contributions are the integrals along the vertical edges $\Gamma_1, \Gamma_3$, i.e., $M_1, M_3$.[4] We now estimate $M_1$. Using the submultiplicative property of the spectral norm and factoring out $E$, we have

$$M_1 \leq \int_{\Gamma_1} \frac{1}{|z|} \cdot \|(zI-A)^{-1}\| \cdot \|E\| \cdot \|(zI-A)^{-1}\| \, |dz| = \int_{\Gamma_1} \|E\| \cdot \frac{1}{|z| \cdot \min_{1 \leq i \leq n} |z-\lambda_i|^2} |dz|.$$

Here, we use the standard identity that

$$\|(zI-A)^{-1}\| = \frac{1}{\min_{1 \leq i \leq n} |z-\lambda_i|}.$$

The key observation is that $|z - \lambda_i| \geq |z - \lambda_n|$ for any $z \in \Gamma_1$ and $1 \leq i \leq n$. Hence, the r.h.s is at most

$$\int_{\Gamma_1} \|E\| \cdot \frac{1}{|z| \cdot |z-\lambda_n|^2} |dz|.$$

By the definition of $\Gamma_1 := \{(x_0, t) | -T \leq t \leq T\}$, we parameterize $z = x_0 + \mathbf{i}t$ for $t \in [-T, T]$. Then $|z| = \sqrt{x_0^2 + t^2}$ and $|z - \lambda_n|^2 = (\lambda_n - x_0)^2 + t^2$, since $\lambda_n$ is real. Moreover, on this segment, $|dz| = dt$. Therefore, the integral becomes

$$\|E\| \cdot \int_{-T}^{T} \frac{1}{\sqrt{x_0^2+t^2}((x_0-\lambda_n)^2+t^2)} dt = \|E\| \cdot \int_{-T}^{T} \frac{1}{((\lambda_n/2)^2+t^2)^{3/2}} dt.$$

By Lemma C.2,

$$\int_{-T}^{T} \frac{1}{((\lambda_n/2)^2+t^2)^{3/2}} dt \leq \frac{\pi}{(\lambda_n/2)^2} = \frac{4\pi}{\lambda_n^2}.$$

Therefore, $M_1 \leq \frac{4\pi\|E\|}{\lambda_n^2}$.

In a similar manner, replace $\Gamma_1$ by $\Gamma_3 := \{(x_1, t) | -T \leq t \leq T\}$, we also obtain

$$M_3 \leq \int_{\Gamma_3} \frac{\|E\| |dz|}{|z| \cdot \min_{1 \leq i \leq n} |z-\lambda_i|^2} \leq \int_{-T}^{T} \frac{\|E\| dt}{\sqrt{x_1^2+t^2}((x_1-\lambda_{n-p})^2+t^2)}.$$

By Lemma C.2 and the fact that $|x_1 - \lambda_{n-p}| = \delta_{n-p}/2$, $M_3$ is at most

$$\frac{\pi\|E\|}{x_1 \cdot \delta_{n-p}/2} = \frac{4\pi\|E\|}{(\lambda_{n-p}+\lambda_{n-p+1})\delta_{n-p}} \leq \frac{4\pi\|E\|}{\lambda_{n-p}\delta_{n-p}}.$$

Arguing similarly, we also obtain that $M_2, M_4 \leq \frac{\|E\|}{4\lambda_1^2}$ (Section C.2), and hence $M_2 + M_4 < \|E\|/(2\lambda_1^2)$. These estimates imply

$$F_1 \leq \frac{1}{2\pi}(M_1 + M_2 + M_3 + M_4) \leq \frac{2\|E\|}{\lambda_n^2} + \frac{2.5\|E\|}{\lambda_{n-p}\delta_{n-p}}.$$

The last inequality follows the facts that $\lambda_n \leq \lambda_{n-p}, \lambda_{n-p+1}$, and $\max\{\lambda_{n-p}\delta_{n-p}, \lambda_n^2\} < \lambda_1^2$. This $F_1$'s upper bound and Lemma 3.1 prove Theorem 2.1.

**Proving the contour bootstrapping lemma (Lemma 3.1).** The first observation is that using the Sherman-Morrison-Woodbury formula $M^{-1} - (M+N)^{-1} = (M+N)^{-1}NM^{-1}$ [28] and the fact that $\tilde{A} = A + E$, we obtain

$$(zI-A)^{-1} - (zI-\tilde{A})^{-1} = (zI-A)^{-1}E(zI-\tilde{A})^{-1}.$$

Using this, we can rewrite

$$F = \frac{1}{2\pi} \int_{\Gamma} \|z^{-1}(zI-A)^{-1}E(zI-\tilde{A})^{-1}\| |dz|$$

as

$$\frac{1}{2\pi} \int_{\Gamma} \|f(z)(zI-A)^{-1}E(zI-A)^{-1} - f(z)(zI-A)^{-1}E[(zI-A)^{-1} - (zI-\tilde{A})^{-1}]\| |dz|.$$

---

[4]In [53], the contour construction was free to extend rightward, making the primary contribution to the integral only come from the left vertical segment. In contrast, our contour has to be more restrictive, and both vertical segments play an equally essential role in the analysis.

Using the triangle inequality, we first see that $F$ is at most

$$\frac{1}{2\pi} \int_\Gamma \|z^{-1}(zI - A)^{-1}E(zI - A)^{-1}\| |dz| + \underbrace{\frac{1}{2\pi} \int_\Gamma \|z^{-1}(zI - A)^{-1}E[(zI - A)^{-1} - (zI - \tilde{A})^{-1}]\| |dz|}.$$

Next is the key observation that the second term in the equation above can be rearranged and upper-bounded as follows, so that the original perturbation appears again:

$$\frac{\max_{z \in \Gamma} \|(zI - A)^{-1}E\|}{2\pi} \int_\Gamma \|z^{-1}[(zI - A)^{-1} - (zI - \tilde{A})^{-1}]\| |dz|.$$

Thus, we have

$$F \leq F_1 + \max_{z \in \Gamma} \|(zI - A)^{-1}E\| \cdot F.$$

Furthermore, our assumption that $4\|E\| \leq \min\{\delta_{n-p}, \lambda_n\}$ and the definition of $\Gamma$ imply

$$\max_{z \in \Gamma} \|(zI - A)^{-1}E\| \leq \|E\| \cdot \max_{z \in \Gamma} \|(zI - A)^{-1}\| = \frac{\|E\|}{\min_{z \in \Gamma, i \in [n]} |z - \lambda_i|} = \frac{\|E\|}{\min\{\delta_{n-p}, \lambda_n\}/2} \leq \frac{1}{2}.$$

Equivalently, $F \leq F_1 + F/2$, and hence $F \leq 2F_1$. We thus complete the proof overview of Lemma 3.1 and, consequently, Theorem 2.1.

# 4 Empirical results

We empirically validate the perturbation bound in Theorem 2.1, demonstrating that it

(i) holds on real datasets, and

(ii) yields significantly tighter estimates than the EYM–N bound.

## 4.1 Assumption of Theorem 2.1 on real-world datasets

Theorem 2.1 requires the spectral condition $4\|E\| < \min\{\lambda_n, \delta_{n-p}\}$, where $\lambda_n$ is the smallest eigenvalue of the matrix $A$, and $\delta_{n-p} := \lambda_{n-p} - \lambda_{n-p+1}$ denotes the eigengap near the truncation threshold. We translate this requirement into a data-dependent upper bound on the noise variance for two real-world matrices.

We perform this analysis on two matrices: the 1990 US Census covariance matrix ($n = 69$) and the BCSSTK09 stiffness matrix ($n = 1083$). Specifically, we first compute $\lambda_n$ and $\delta_{n-p}$ for the smallest $p$ such that the spectral tail satisfies

$$\frac{\|A^{-1} - A_p^{-1}\|}{\|A^{-1}\|} < 0.05,$$

ensuring that at least 95% of the inverse spectral mass is retained. We then translate these spectral quantities into the maximum permissible noise level $\|E\|$, and derive the corresponding sub-Gaussian variance threshold

$$\Delta^{\max} := \frac{\min\{\lambda_n, \delta_{n-p}\}}{8\sqrt{n}}.$$

**Datasets.** We use two widely studied matrices: the $69 \times 69$ US Census covariance matrix from the UCI ML repository [5], commonly used in studies on differentially private PCA [4, 12, 36], and the $1083 \times 1083$ BCSSTK09 matrix [15], a stiffness matrix arising from a finite-element model of a clamped plate [6, 9, 11, 17, 50].

**Noise model and variance threshold.** We consider symmetric noise matrices $E$ with i.i.d. sub-Gaussian entries (mean zero, variance proxy $\Delta^2$). With high probability, $\|E\| = (2 + o(1))\Delta\sqrt{n}$, as established in [58, 60]. Thus, Theorem 2.1 is valid whenever $4(2 + o(1))\Delta\sqrt{n} < \min\{\lambda_n, \delta_{n-p}\}$, equivalently, $\Delta < \Delta^{\max}$.

**Results and conclusion.** For the US Census matrix with $p = 17$, we compute $\Delta^{\max} \approx 47.8$; for BCSSTK09 with $p = 8$, we find $\Delta^{\max} \approx 26.9$. These thresholds comfortably exceed the noise levels commonly used in practice. For instance, in differential privacy, Laplacian noise with scale $b$ satisfies $\|E\| \leq \sqrt{2}b$. Since $\varepsilon$-DP corresponds to $b = 1/\varepsilon$, Theorem 2.1 applies as long as $\varepsilon > 0.03$—well within the commonly accepted range for strong privacy [40]. Similarly, prior work using BCSSTK09 applies noise at the level $\|E\| < 10^{-5}\|A\| \approx 6.7 \times 10^2$, which translates to $\Delta \approx 10.2 < \Delta^{\max}$.

Section F (Table 1) confirms that this safety margin persists for a range of $p$ values. We conclude that the assumptions of Theorem 2.1 are satisfied in several practical settings, making it broadly applicable to workflows in differential privacy, structural engineering, and numerical linear algebra.

## 4.2 Empirical sharpness of Theorem 2.1

To gauge the practical sharpness of our new low-rank inverse–perturbation bound, we benchmark it on three markedly different matrices—a dense covariance matrix from the 1990 US Census, a large sparse stiffness matrix (BCSSTK09), and a synthetic discretized Hamiltonian with an almost linear spectrum. By injecting both Gaussian Orthogonal Ensemble and Rademacher noise at ten escalating levels that respect the stability requirement of Theorem 2.1, we create a broad test bed that spans dense, sparse, and near-Toeplitz spectra as well as moderate to severe perturbations. The goal is to compare

(i) the true error,

(ii) our bound, and

(iii) the Eckart–Young-Mirsky–Neumann (EYM–N) bound under the same conditions.

**Setting.** In this subsection, we consider three different matrices $A$:

(i) real matrices: the $69 \times 69$ covariance of the 1990 US Census ($A :=$ Census, $n = 69$),

(ii) the $1083 \times 1083$ BCSSTK09 stiffness matrix ($A :=$ BCSSTK09, $n = 1083$), and

(iii) synthetic matrix: the approximately linear spectrum $A$ ($A :=$ Discretized Hamiltonian) derived by discretizing the 1–D quantum harmonic oscillator [5] on $n \in \{500, 1000\}$ grid points (see Appendix G for the detailed construction).

We set the low-rank parameter $p$ satisfies $\|A^{-1} - A_p^{-1}\|/\|A^{-1}\| < 0.05$. This yields $p = 17$ for $A =$ Census, $p = 8$ for $A =$ BCSSTK09, and $p = 10$ for $A =$ Discretized Hamiltonian.

We perturb each $A$ by either Gaussian Orthogonal Ensemble (GOE) noise $E_1$ or Rademacher noise $E_2$. Each $E_k$ is scaled by ten equally spaced factors $C_A$ so that $4C_A\|E_k\|$ spans up to $\min\{\lambda_n, \delta_{n-p}\}$, i.e., $C_A \in \{1.5, 2.0, \ldots, 6\}$ for $A =$ Census, $C_A \in \{1.2, 1.4, \ldots, 3\}$ for $A =$ BCSSTK09, and $C_A \in \{10^{-4}, 10^{-3.67}, \ldots, 10^{-1}\}$ for $A =$ Discretized Hamiltonian; see Table 1 and Appendix G. This scaling range ensures that the assumption of Theorem 2.1 is satisfied.

**Evaluation.** For each configuration $(A, E_k, n, p)$, we report:

(i) the empirical error $\|(\tilde{A}^{-1})_p - A_p^{-1}\|$ (100 trials),

(ii) our bound $\frac{4\|E\|}{\lambda_n^2} + \frac{5\|E\|}{\lambda_{n-p}\delta_{n-p}}$, and

(iii) the EYM–N bound $\frac{8\|E\|}{3\lambda_n^2} + \frac{2}{\lambda_{n-p}}$.

For the 1990 US Census, we additionally preprocess the data to ensure all entries are numeric: we discard the header row and the indexing column, then replace every non-numeric field with 0. We record the ratio $\frac{our\ bound}{actual\ error}$. As is standard, all numerical results are reported as mean $\pm$ standard deviation in .4e format, and the curves for *Actual Error*, *Our Bound*, and *EYM–N Bound* are plotted with error bars (cap width = 3pt) and logarithmic $y$-axis.

**Results and conclusion.** For every matrix tested-the $69 \times 69$ US-Census covariance, the $1083 \times 1083$ BCSSTK09 stiffness matrix, and the discretised Hamiltonians with $n \in \{500, 1000\}$—our low-rank inverse bound consistently outperforms the EYM–N estimate and closely follows the measured error for all noise models $E_k$ and scaling factors $C_A$; see Figures 1–2. (The error bars for *Our Bound* and the *EYM–N Bound* are too small to discern.) In every experiment $\frac{our\ bound}{actual\ error} < 10$, whereas the EYM–N bound is typically looser by more than an order of magnitude; see Tables 2-9. This improvement is uniform across matrix sizes $n \in \{69, 500, 1000, 1083\}$, demonstrating that our estimate captures the leading error term in practice and is therefore a reliable error certificate for low-rank inverse approximations.

---

[5]The inverse harmonic oscillator and its discretized version are central to many studies in spectral perturbation theory; e.g., implicit time-stepping, preconditioning in quantum simulations, and the design of Gaussian-process covariance kernels [7, 34, 49, 51]. Low-rank approximations of these inverses enable fast $O(n \log n)$ solvers and reduced-order models [24, 38, 55].

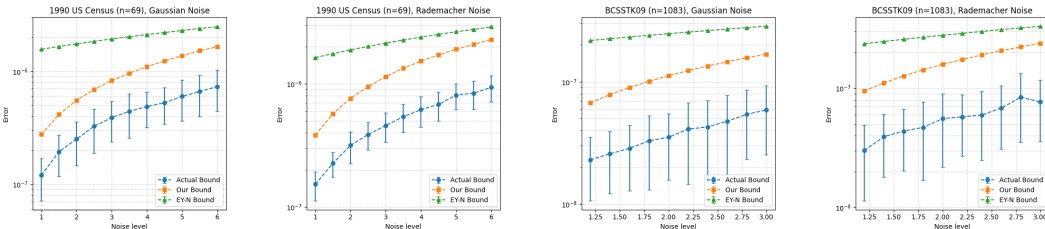

Figure 1: Four panels show (i) Actual Error, (ii) Our Bound, and (iii) EYM–N Bound, over 100 trials for real-world matrices $A =$ Census ($n = 69$, $p = 17$) and $A =$ BCSSTK09 ($n = 1083$, $p = 8$), perturbed by Gaussian/Rademacher noise.

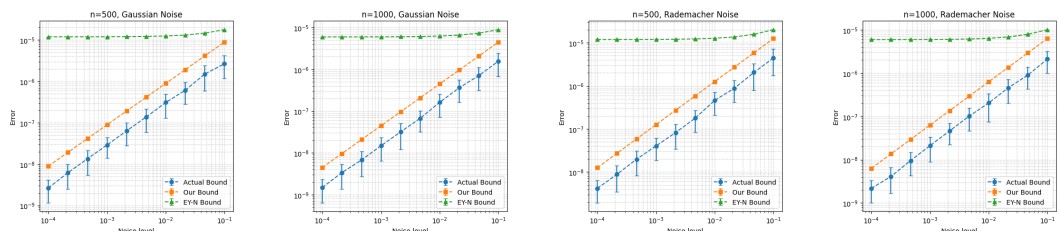

Figure 2: Four panels show (i) Actual Error, (ii) Our Bound, and (iii) EYM–N Bound, over 100 trials for $A =$ Hamiltonian ($p = 10$, $n \in \{500, 1000\}$) perturbed by Gaussian/Rademacher noise.

# 5 Conclusion, limitations, and future work

We present the first non-asymptotic spectral-norm perturbation bounds for low-rank approximations of matrix inverses under general additive noise. Our results characterize how the error $\|(\widetilde{A}^{-1})_p - A_p^{-1}\|$ depends on spectral quantities such as the smallest eigenvalue $\lambda_n$, the eigengap $\delta_{n-p}$, and the alignment of noise with low-curvature eigenspaces. In regimes where these quantities are well-behaved, our bound improves upon classical Neumann-based estimates by up to a $\sqrt{n}$-factor. This analysis introduces a new application of contour bootstrapping to the non-entire function $f(z) = 1/z$, allowing us to isolate and control the impact of perturbations on inverse approximations projected onto the smallest eigencomponents of $A$.

We validate our bounds on diverse matrix classes—including dense covariance matrices, sparse stiffness matrices, and discretized quantum Hamiltonians—under both Gaussian and Rademacher noise. Across all settings, our bound tracks the empirical error within a small constant factor and consistently outperforms the Eckart–Young-Mirsky–Neumann baseline, often by over an order of magnitude. These findings yield robust, spectrum-aware guarantees for low-rank inverse estimation in noisy numerical pipelines.

Despite these contributions, several limitations remain. Our guarantees depend on spectral quantities that may be difficult to estimate efficiently, especially in black-box or data-driven scenarios. In particular, verifying the gap condition $\delta_{n-p} > 4\|E\|$ requires accurate access to the tail of the spectrum, which can be computationally demanding. Moreover, our results are tailored to static matrices and do not directly extend to adaptive or iterative settings where the matrix evolves over time.

Nonetheless, our framework provides a principled tool for certifying the stability of inverse-based methods in the presence of noise. In optimization and machine learning, it can inform the use of low-rank Hessian approximations, preconditioners, or trust-region updates under noisy curvature information. Future directions include analyzing structured or time-varying noise, developing adaptive gap estimators, obtaining the sharp perturbation bounds of low-rank inverse approximation for other structured metrics such as Schatten-$p$ norm or the Ky Fan norm, and extending contour techniques to other non-entire matrix functions such as resolvents or matrix roots.

## Acknowledgments

This work was funded in part by NSF Award CCF-2112665, Simons Foundation Award SFI-MPS-SFM-00006506, and NSF Grant AWD 0010308.

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

# Contents

## A  Extension of Theorem 2.1 for an arbitrary symmetric matrix A

In this section, we extend Theorem 2.1 to the perturbation of the best rank-$p$ approximation of the inverse when $A$ is a symmetric matrix. To simplify the presentation, we assume that the eigenvalues (singular values) are different, so the eigenvectors (singular vectors) are well-defined (up to signs). However, our results hold for matrices with multiple eigenvalues.

For a general symmetric matrix $A$, we interpret its collection of $p$-least singular values as follows. Since $-1 < 0$, to make the perturbation well-defined, there exists a natural number $1 \leq k \leq n$ such that

$$\lambda_1 > \lambda_2 > \cdots > \lambda_k > 0 > \lambda_{k+1} > \cdots > \lambda_n.$$

Hence, there is an integer number $k_1$ such that $\{\sigma_n, \sigma_{n-1}, \ldots, \sigma_{n-p+1}\} \equiv \{|\lambda_i|, i \in S\}$ for

$$S := \{k - (k_1 - 1), k - (k_1 - 2), \ldots, k, k+1, k+2, \ldots, k + (p - k_1)\}.$$

In general, there is a permutation $\pi$ of $[n]$ such that $\sigma_{n-p} = |\lambda_{\pi(p)}|$ for all $0 \leq p \leq n - 1$. We have the following extension of Theorem 2.1

**Theorem A.1.** *If* $4\|E\| < \min\{\delta_{k-k_1}, \delta_{k+p-k_1}, \sigma_n\}$, *and* $\sigma_{n-p} - \sigma_{n-p+1} > 2\|E\|$ *then*

$$\|(\tilde{A}^{-1})_p - A_p^{-1}\| \leq \frac{4\|E\|}{\lambda_k^2} + \frac{5\|E\|}{\lambda_{k-k_1}\delta_{k-k_1}} + \frac{4\|E\|}{|\lambda_{k+1}|^2} + \frac{5\|E\|}{|\lambda_{k+p-k_1+1}|\delta_{k+p-k_1}}.$$

Note that when $A$ is not PD, $\tilde{A}$ with the eigenvalues $\tilde{\lambda}_1 \geq \tilde{\lambda}_2 \geq \cdots \geq \tilde{\lambda}_n$ is not necessarily PD. And hence, the set $\{|\tilde{\lambda}_{k-k_1+1}|, \ldots, |\tilde{\lambda}_k|, |\tilde{\lambda}_{k+1}|, \ldots, |\tilde{\lambda}_{k+p-k_1}|\}$ may not correspond to the $p$ least singular values of $\tilde{A}$. This issue is resolved by enforcing the singular-value gap condition $\sigma_{n-p} - \sigma_{n-p+1} > 2\|E\|$.

**Remark A.2.** *This extension is important when $A$ has both positive and negative eigenvalues. In real-world applications where data is often arbitrary, it is natural for the eigenvalues of $A$ to span both signs. While singular value decomposition (SVD) could be used to apply Theorem 2.1, singular value gaps are typically small. By working directly with eigenvalues, we exploit the fact that the eigenvalue gaps $\delta_{k-k_1} = \lambda_{k-k_1} - \lambda_{k-k_1+1}$ and $\delta_{k+(p-k_1)} = \lambda_{k+p-k_1} - \lambda_{k+p-k_1+1}$ are significantly larger than $\sigma_{n-p} - \sigma_{n-p+1}$ when $\lambda_{\pi(p)} \cdot \lambda_{\pi(p+1)} < 0$. For example, if $\lambda_{\pi(n-p)} = -\sqrt{n} + \log n, \lambda_{\pi(n-p+1)} = \sqrt{n}, \lambda_{\pi(n-p+2)} = -2\sqrt{n}, \lambda_{\pi(n-p+3)} = 2\sqrt{n} + \log n$, then*

$$\min\{\delta_{k-k_1}, \delta_{k+p-k_1}\} = \Theta(\sqrt{n}) \text{ while } \sigma_{n-p} - \sigma_{n-p+1} = \log n.$$

**Proof of Theorem A.1**  Since the spectrum of $A$ is

$$\lambda_1 \geq \lambda_2 \geq \cdots \geq \lambda_k > 0 > \lambda_{k+1} \geq \cdots \geq \lambda_n,$$

the spectrum of $A^{-1}$ is

$$\lambda_k^{-1} \geq \lambda_{k-1}^{-1} \geq \cdots \geq \lambda_1^{-1} > 0 > \lambda_n^{-1} \geq \lambda_{n-1}^{-1} \geq \cdots \geq \lambda_{k+1}^{-1}.$$

We construct the contour $\Gamma$ as follows:

$$\Gamma = \Gamma^{[1]} \cup \Gamma^{[2]} \cup L,$$

in which $\Gamma^{[1]}$ and $\Gamma^{[2]}$ are rectangles, whose vertices are

$\Gamma^{[1]} : (a_0, T), (a_1, T), (a_1, -T), (a_0, -T)$ with $a_0 := \lambda_k/2, a_1 := \lambda_{k-(k_1-1)} + \delta_{k-k_1}/2, T := 2\sigma_1$;

and

$\Gamma^{[2]} : (b_0, T), (b_1, T), (b_1, -T), (b_0, -T)$ with $b_0 := \lambda_{k+1}/2, b_1 := \lambda_{k+p-k_1} - \delta_{k+p-k_1}/2, T := 2\sigma_1$;

and $L$ is a segment, connecting $(b_0, T)$ and $(a_0, T)$.

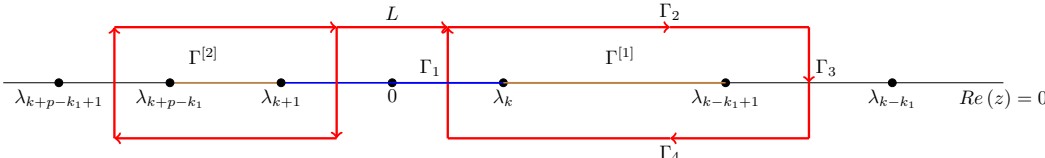

Applying the contour bootstrapping argument, we obtain

$$\left\| (\tilde{A}^{-1})_p - A_p^{-1} \right\| \leq 2F_1 := \frac{1}{\pi} \int_\Gamma \left\| z^{-1}(zI - A)^{-1} E(zI - A)^{-1} \right\| |dz|$$

$$= 2(F_1^{[1]} + F_1^{[2]} + F_1^{[L]}),$$

(3)

in which

$$F_1^{[1]} := \frac{1}{2\pi} \int_{\Gamma^{[1]}} \left\| z^{-1}(zI - A)^{-1} E(zI - A)^{-1} \right\| |dz|,$$

$$F_1^{[2]} := \frac{1}{2\pi} \int_{\Gamma^{[2]}} \left\| z^{-1}(zI - A)^{-1} E(zI - A)^{-1} \right\| |dz|$$

$$F_1^{[L]} := \frac{1}{2\pi} \int_L \left\| z^{-1}(zI - A)^{-1} E(zI - A)^{-1} \right\| |dz|.$$

Now, we are going to bound $F_1^{[1]}$. First, we split $\Gamma^{[1]}$ into four segments:

- $\Gamma_1 := \{(a_0, t) | - T \leq t \leq T\}$.
- $\Gamma_2 := \{(x, T) | a_0 \leq x \leq a_1\}$.
- $\Gamma_3 := \{(a_1, t) | T \geq t \geq -T\}$.
- $\Gamma_4 := \{(x, -T) | a_1 \geq x \geq a_0\}$.

Therefore,

$$F_1^{[1]} = \sum_{l=1}^4 \frac{1}{2\pi} \int_{\Gamma_l} \left\| z^{-1}(zI - A)^{-1} E(zI - A)^{-1} \right\| |dz|.$$

Notice that

$$\left\| z^{-1}(zI - A)^{-1} E(zI - A)^{-1} \right\| \leq \|E\| \frac{1}{|z| \times \min_{i \in [n]} |z - \lambda_i|^2},$$

we further obtain

$$2\pi F_1^{[1]} \leq M_1 + \|E\| (N_2 + N_4) + M_3,$$

in which

$$M_1 := \int_{\Gamma_1} \left\| z^{-1}(zI - A)^{-1} E(zI - A)^{-1} \right\| |dz|,$$

$$M_3 := \int_{\Gamma_3} \left\| z^{-1}(zI - A)^{-1} E(zI - A)^{-1} \right\| |dz|,$$

and

$$N_l := \int_{\Gamma_l} \frac{1}{|z| \times \min_{i \in [n]} |z - \lambda_i|^2} |dz| \text{ for } l \in \{2, 4\}.$$

We use the following lemmas (their proofs are delayed to Section C.1 and Section C.2).

**Lemma A.3.** *Under assumptions of Theorem A.1,*

$$M_1 \leq \frac{4\pi \|E\|}{\lambda_k^2}.$$

**Lemma A.4.** *Under assumptions of Theorem A.1,*

$$M_3 \leq \min \frac{4\pi \|E\|}{\lambda_{k-k_1} \delta_{k-k_1}}.$$

**Lemma A.5.** *Under assumptions of Theorem A.1,*

$$N_2, N_4 \leq \frac{1}{T^2}.$$

Together Lemma A.3, Lemma A.4, and Lemma A.5 imply

$$F_1^{[1]} \leq \frac{1}{2\pi} \left( M_1 + M_3 + \frac{\|E\|}{N_2} + \frac{\|E\|}{N_4} \right) \leq \frac{2\|E\|}{\lambda_k^2} + \frac{2\|E\|}{\lambda_{k-k_1} \delta_{k-k_1}} + \frac{\|E\|}{4\pi \sigma_1^2}.$$

By a similar manner, we also obtain

$$F_1^{[2]} \leq \frac{2\|E\|}{\lambda_{k+1}^2} + \frac{2\|E\|}{|\lambda_{k+p-k_1+1}| \delta_{k+p-k_1}} + \frac{\|E\|}{4\pi \sigma_1^2}.$$

For bounding $F_1^{[L]}$, we use the following lemma, whose proof is also delayed to Section C.3.

**Lemma A.6.** *Under assumptions of Theorem A.1,*

$$F_1^{[L]} \leq \frac{1}{2\pi} \frac{|a_0 - b_0| \times \|E\|}{T^3}.$$

Since $|a_0 - b_0| \leq |a_0| + |b_0| \leq 2\sigma_1 = T$, we further obtain $F_1^{[L]} \leq \frac{\|E\|}{8\pi\sigma_1^2}$. Combining above estimates for $F_1^{[1]}, F_1^{[2]}$, and $F_1^{[L]}$, we finally obtain

$$F \leq 2F_1 \leq \frac{4\|E\|}{\lambda_k^2} + \frac{4\|E\|}{\lambda_{k-k_1}\delta_{k-k_1}} + \frac{\|E\|}{\pi\sigma_1^2} + \frac{4\|E\|}{\lambda_{k+1}^2} + \frac{4\|E\|}{|\lambda_{k+p-k_1+1}|\delta_{k+p-k_1}},$$

which is less than the r.h.s of Theorem A.1. We complete the proof.

**Remark A.7.** *Our approach directly extends to the case where $A$ is a symmetric PSD of rank $r < n$ (rank-deficient). In this setting, one replaces $A^{-1}$, $A_p^{-1}$, and $\tilde{A}_p^{-1}$ with $A^\dagger$ (the pseudoinverse of $A$), $A_p^\dagger$, and the projection of $\tilde{A}^\dagger$ onto the subspace corresponding to the nonzero eigenvalues $\tilde{\lambda}_r, \tilde{\lambda}_{r-1}, \ldots, \tilde{\lambda}_{r-p+1}$ respectively. The contour $\Gamma$ can then be constructed with respect to $(\lambda_r, \lambda_{r-p+1}, \delta_{r-p})$, and the analysis proceeds similarly.*

# B    Refinements of Theorem 2.1 and Theorem A.1

By looking at the finer structure of $M_1$ and $M_3$, one can obtain a more nuanced bound. The key idea is to control the spectral decay of $A$ and to take into account the interaction between $E$ and the $p$-bottom eigenvectors of $A$. Recall the notations from the previous section. Given the eigenvalues of $A$, $\lambda_1 > \lambda_2 > \cdots > \lambda_k > 0 > \lambda_{k+1} > \cdots > \lambda_n$, there is an integer number $k_1$ such that $\{\sigma_n, \sigma_{n-1}, \ldots, \sigma_{n-p+1}\} \equiv \{|\lambda_i|, i \in S\}$ for

$$S := \{k - (k_1 - 1), k - (k_1 - 2), \ldots, k, k+1, k+2, \ldots, k + (p - k_1)\}.$$

In general, there is a permutation $\pi$ of $[n]$ such that $\sigma_{n-p} = |\lambda_{\pi(p)}|$ for all $0 \leq p \leq n - 1$.

To characterize how quickly the singular values of $A$ grow, we define the *doubling distance* $r \geq p$ (with respect to the index $p$) as follows. Let $k - (k_1 - 1) \leq r_1 \leq k - 1$ be the smallest integer such that $2\lambda_k \leq \lambda_{k-r_1}$ and let $1 \leq r_2 \leq n - k$ be the smallest integer such that $2|\lambda_{k+1}| \leq |\lambda_{k+r_2+1}|$. Set $r := \min\{r_1, r_2\}$ if $1 \leq k \leq n - 1$ and $r := \max\{r_1, r_2\}$ if $k \in \{0, n\}$. Define the *important subset* $I_r := \{i \mid k + r_2 \geq i \geq k - r_1 + 1\}$ and *the interaction parameter* $x := \max_{i,j \in I_r} |u_i^\top E u_j|$. The asymptotic refinement of Theorem A.1 is

**Theorem B.1.** *If $4\|E\| < \delta := \min\{\delta_{k-k_1}, \delta_{k+p-k_1}, \sigma_n\}$ and $\sigma_{n-p} - \sigma_{n-p+1} > 2\|E\|$, then*

$$\|(\tilde{A}^{-1})_p - A_p^{-1}\| \leq O\left(\frac{\|E\|}{\sigma_n\sigma_{n-r}} + \frac{r^2x}{\lambda_k^2} + \frac{r^2x}{\lambda_{k-k_1}\delta_{k-k_1}} + \frac{r^2x}{|\lambda_{k+1}|^2} + \frac{r^2x}{|\lambda_{k+p-k_1+1}|\delta_{k+p-k_1}}\right).$$

In particular, when $A$ is PD, the *doubling distance* $r \geq p$ is simply the smallest positive integer satisfying $2\lambda_{n-p+1} \leq \lambda_{n-r}$, and the *interaction term* is $x := \max_{n-r+1 \leq i, j \leq n} |u_i^\top E u_j|$. The refinement of Theorem 2.1 is

**Theorem B.2** (**Refinement of Theorem 2.1**)**.** *If $4\|E\| < \min\{\delta_{n-p}, \lambda_n\}$, then*

$$\|(\tilde{A}^{-1})_p - A_p^{-1}\| \leq O\left(\frac{\|E\|}{\lambda_n\lambda_{n-r}} + \frac{r^2x}{\lambda_n^2} + \frac{r^2x}{\lambda_{n-p}\delta_{n-p}}\right).$$

**Comparison to EYM–N bound in the intermediate regime** $\min\{\lambda_n, \delta_{n-p}\} \gg \|E\| \gg \frac{\lambda_n^2}{\lambda_{n-p}}$ When $A$ is PD and the noise $E$ is in the intermediate regime, our result improves upon the classical bound by a factor of $O\left(\min\left\{\frac{\lambda_{n-r}}{\lambda_n}, \frac{\|E\|}{r^2x}\right\}\right)$. When $E$ is a Wigner random noise, with high probability, $\|E\| = (2 + o(1))\sqrt{n}$ and $x = O(\log n)$ [60, 41, 54], this gaining factor simplifies to $\min\left\{\frac{\lambda_{n-r}}{O(\lambda_n)}, \frac{\sqrt{n}}{O(r^2 \log n)}\right\}$, yielding up to a $\sqrt{n}$-factor improvement. This is achievable when the stable rank $\text{sr}(A^{-1})$ is $\sim \tilde{O}(1)$ [6] and $E$ is a Wigner random noise [41, 54]. As a concrete example, consider $A$ with spectrum $\{(1 - \frac{1}{2})n, \ldots, (1 - \frac{1}{n-9})n, 40\sqrt{n}, 8\sqrt{n}, 4\sqrt{n}\}$, perturbed by the standard Gaussian noise $E \sim \mathcal{N}(0, I)$. For $p = 6$ ( $r_p = 11$), the EYM–N bound gives $O(n^{-1/2})$, whereas Theorem 2.1 yields $\tilde{O}(n^{-1})$, a clear $\sqrt{n}$-level gain.

---

[6] $\tilde{O}$ hides poly-log factors

**Proof of Theorem B.1** This proof follows exactly the proof of Theorem A.1, except the following asymptotically finer estimates of $M_1$ and $M_3$.

**Lemma B.3.** *Under the assumption of Theorem B.1,*

$$M_1 \leq O\left(\frac{\|E\|}{\lambda_k \sigma_{n-r}} + \frac{r^2 x}{\lambda_k^2}\right).$$

**Lemma B.4.** *Under the assumption of Theorem B.1,*

$$M_3 \leq O\left(\frac{\|E\|}{\lambda_{k-k_1} \sigma_{n-r}} + \frac{r^2 x}{\lambda_{k-k_1} \delta_{k-k_1}}\right).$$

The proofs of these lemmas will be delayed to the next section. Combining the estimates of $N_2, N_4$ from Lemma A.5 with Lemma B.3 and Lemma B.4, we obtain $F_1^{[1]}$ at most

$$O\left(\frac{\|E\|}{\lambda_k \sigma_{n-r}} + \frac{r^2 x}{\lambda_k^2} + \frac{\|E\|}{\lambda_{k-k_1} \sigma_{n-r}} + \frac{r^2 x}{\lambda_{k-k_1} \delta_{k-k_1}} + \frac{\|E\|}{\sigma_1^2}\right) \leq O\left(\frac{\|E\|}{\sigma_n \sigma_{n-r}} + \frac{r^2 x}{\lambda_k^2} + \frac{r^2 x}{\lambda_{k-k_1} \delta_{k-k_1}}\right).$$

In a similar manner, we also obtain

$$F_2^{[1]} \leq O\left(\frac{\|E\|}{\sigma_n \sigma_{n-r}} + \frac{r^2 x}{|\lambda_{k+1}|^2} + \frac{r^2 x}{|\lambda_{k+p-k_1+1}| \delta_{k+p-k_1}}\right).$$

Combining these estimates with Lemma A.6 that $F_1^{[L]} = O\left(\|E\|/\sigma_1^2\right)$, we finally obtain

$$F \leq 2F_1 \leq O\left(\frac{\|E\|}{\sigma_n \sigma_{n-r}} + \frac{r^2 x}{\lambda_k^2} + \frac{r^2 x}{\lambda_{k-k_1} \delta_{k-k_1}} + \frac{\|E\|}{\sigma_n \sigma_{n-r}} + \frac{r^2 x}{|\lambda_{k+1}|^2} + \frac{r^2 x}{|\lambda_{k+p-k_1+1}| \delta_{k+p-k_1}} + \frac{\|E\|}{\sigma_1^2}\right),$$

which simplifies to

$$O\left(\frac{\|E\|}{\sigma_n \sigma_{n-r}} + \frac{r^2 x}{\lambda_k^2} + \frac{r^2 x}{\lambda_{k-k_1} \delta_{k-k_1}} + \frac{r^2 x}{|\lambda_{k+1}|^2} + \frac{r^2 x}{|\lambda_{k+p-k_1+1}| \delta_{k+p-k_1}}\right).$$

We complete the proof.

## C  Contour integral estimations

In this section, we present the contour integral estimations used in the previous section: Lemma A.3, Lemma B.3, Lemma A.4, Lemma B.4 (integration over vertical segments); Lemma A.5 (integration over horizontal segments), and Lemma A.6 (integration over L-segment). We first present two technical lemmas, which are used several times in the upcoming sections.

**Lemma C.1.** *Let $a, T$ be positive numbers such that $a \leq T$. Then,*

$$\int_{-T}^{T} \frac{1}{t^2 + a^2} dt \leq \frac{\pi}{a}.$$

**Proof of Lemma C.1** We have

$$\begin{aligned}
\int_{-T}^{T} \frac{1}{t^2 + a^2} dt &= 2 \int_0^T \frac{1}{t^2 + a^2} \\
&= \frac{2}{a} \arctan(T/a) \\
&\leq \frac{2}{a} \cdot \frac{\pi}{2} = \frac{\pi}{a}.
\end{aligned}$$

**Lemma C.2.** *Let $a, b, c, T$ be positive numbers such that $a, b, c \leq T$. Then,*

$$\int_{-T}^{T} \frac{1}{\sqrt{(t^2 + a^2)(t^2 + b^2)(t^2 + c^2)}} dt \leq \frac{\pi}{\max\{a,b,c\} \times \min\{a,b,c\}}.$$

**Proof of Lemma C.2** Without the loss of generality, we can assume that $a \leq b \leq c \leq T$. We have

$$\begin{aligned}
\int_{-T}^{T} \frac{1}{\sqrt{(t^2 + a^2)(t^2 + b^2)(t^2 + c^2)}} dt &\leq \frac{1}{c} \cdot \int_{-T}^{T} \frac{1}{\sqrt{(t^2 + b^2)(t^2 + a^2)}} dt \\
&\leq \frac{1}{c} \cdot \int_{-T}^{T} \frac{1}{t^2 + a^2} dt \\
&\leq \frac{\pi}{ac} \quad \text{(by Lemma C.1)}.
\end{aligned}$$

## C.1 Estimating integrals over vertical segments

In this section, we are going estimate $M_1$ - integral over the left vertical segment (prove Lemma A.3 and Lemma B.3), and then by a similar argument, we obtain the upper bound for $M_3$ - integral over the right vertical segment of $\Gamma_1$ (Lemma A.4 and Lemma B.4). First, we estimate $M_1$ as follows.

Using the spectral decomposition $(zI - A)^{-1} = \sum_{i=1}^{n} \frac{u_i u_i^\top}{(z - \lambda_i)}$, we can rewrite $M_1$ as

$$M_1 = \int_{\Gamma_1} \left\| \sum_{n \geq i,j \geq 1} \frac{1}{z(z - \lambda_i)(z - \lambda_j)} u_i u_i^\top E u_j u_j^\top \right\| |dz|.$$

Define a set of indices

$$I_r := \{i \mid k + r_2 \geq i \geq k - r_1 + 1\},$$

and denote its complement is $I_r^c := [n] \setminus I_r$. By the triangle inequality, $M_1$ is at most

$$\int_{\Gamma_1} \left\| \sum_{i,j \in I_r} \frac{1}{z(z - \lambda_i)(z - \lambda_j)} u_i u_i^\top E u_j u_j^\top \right\| |dz| + \int_{\Gamma_1} \left\| \sum_{i,j \in I_r^c} \frac{1}{z(z - \lambda_i)(z - \lambda_j)} u_i u_i^\top E u_j u_j^\top \right\| |dz|$$

$$+ \int_{\Gamma_1} \left\| \sum_{\substack{i \in I_r, j \in I_r^c \\ i \in I_r^c, j \in I_r}} \frac{1}{z(z - \lambda_i)(z - \lambda_j)} u_i u_i^\top E u_j u_j^\top \right\| |dz|.$$

Consider the first term, by the triangle inequality, we have

$$\int_{\Gamma_1} \left\| \sum_{i,j \in I_r} \frac{1}{z(z - \lambda_i)(z - \lambda_j)} u_i u_i^\top E u_j u_j^\top \right\| |dz|$$

$$\leq \sum_{i,j \in I_r} \int_{\Gamma_1} \left\| \frac{1}{z(z - \lambda_i)(z - \lambda_j)} u_i u_i^\top E u_j u_j^\top \right\| |dz|$$

$$= \sum_{i,j \in I_r} \int_{\Gamma_1} \frac{|u_i^\top E u_j| \cdot \|u_i u_j^\top\|}{|z||(z - \lambda_i)(z - \lambda_j)|} |dz|$$

$$\leq \sum_{i,j \in I_r} x \int_{-T}^{T} \frac{1}{\sqrt{(a_0^2 + t^2)((a_0 - \lambda_i)^2 + t^2)((a_0 - \lambda_j)^2 + t^2)}} dt.$$

The last inequality follows the facts that $\|u_i u_j^\top\| = 1, \Gamma_1 := \{z \mid z = a_0 + \mathbf{i}t, -T \leq t \leq T\}$ and $x := \max_{i,j \in I_r} |u_i^\top E u_j|$. By the construction of $\Gamma_1$, we have

$$|a_0 - \lambda_i| \geq \frac{\lambda_k}{2} = a_0 \quad \text{for all} \quad 1 \leq i \leq n. \tag{4}$$

Thus, by Lemma C.2, the r.h.s. is at most

$$r^2 x \cdot \frac{\pi}{a_0^2} = \frac{4\pi r^2 x}{\lambda_k^2},$$

or equivalently,

$$\int_{\Gamma_1} \left\| \sum_{i,j \in I_r} \frac{1}{z(z - \lambda_i)(z - \lambda_j)} u_i u_i^\top E u_j u_j^\top \right\| |dz| \leq \frac{4\pi r^2 x}{\lambda_k^2}. \tag{5}$$

Next, we bound the second term as follows

$$\int_{\Gamma_1} \left\| \sum_{i,j \in I_r^c} \frac{1}{z(z - \lambda_i)(z - \lambda_j)} u_i u_i^\top E u_j u_j^\top \right\| |dz|$$

$$= \int_{\Gamma_1} \left\| \frac{1}{z} \left( \sum_{i \in I - r^c} \frac{u_i u_i^\top}{z - \lambda_i} \right) E \left( \sum_{i \in I_r^c} \frac{u_i u_i^\top}{z - \lambda_i} \right) \right\| |dz|$$

$$\leq \int_{\Gamma_1} |z|^{-1} \cdot \left\| \sum_{i \in I_r^c} \frac{u_i u_i^\top}{z - \lambda_i} \right\| \cdot \|E\| \cdot \left\| \sum_{i \in I_r^c} \frac{u_i u_i^\top}{z - \lambda_i} \right\| |dz| \tag{6}$$

$$\leq \|E\| \int_{\Gamma_1} \frac{1}{|z| \min_{i \in I_r^c} |z - \lambda_i|^2} |dz|$$

$$= \|E\| \int_{-T}^{T} \frac{1}{\sqrt{a_0^2 + t^2} \times \min_{i \in I_r^c} [(a_0 - \lambda_i)^2 + t^2]} dt.$$

Moreover, by the construction of $\Gamma_1$ and the definition of $r$,

$$|a_0 - \lambda_i| \geq \min\{|a_0 - \lambda_{k-r_1}|, |a_0 - \lambda_{k+r_2+1}|\} \geq \min\{\frac{\lambda_{k-r_1}}{2}, \frac{|\lambda_{k+r_2+1}|}{2}\} \geq \frac{\sigma_{n-r}}{2}. \tag{7}$$

where the second inequality follows the fact $i \notin I_r$. Thus, by Lemma C.2, the r.h.s. is at most

$$\|E\| \times \frac{\pi}{a_0 \times \sigma_{n-r}/2} = \frac{4\pi \|E\|}{\sigma_{n-r} \lambda_k}.$$

It follows that

$$\int_{\Gamma_1} \left\| \sum_{i,j \in I_r^c} \frac{z}{(z-\lambda_i)(z-\lambda_j)} u_i u_i^\top E u_j u_j^\top \right\| |dz| \le \frac{4\pi \|E\|}{\sigma_{n-r}\lambda_k}. \tag{8}$$

Now we consider the last term:

$$\int_{\Gamma_1} \left\| \sum_{\substack{i \in I_r, j \in I_r^c \\ i \in I_r^c, j \in I_r}} \frac{1}{z(z-\lambda_i)(z-\lambda_j)} u_i u_i^\top E u_j u_j^\top \right\| |dz| \le 2\|E\| \int_{\Gamma_1} \frac{1}{|z| \min_{i \in I_r, j \in I_r^c} |(z-\lambda_i)(z-\lambda_j)|} |dz|.$$

By (7) and (4), the r.h.s. is at most

$$2\|E\| \int_{-T}^{T} \frac{1}{\sqrt{(t^2+a_0^2)(t^2+a_0^2)(t^2+(\sigma_{n-r}/2)^2)}} dt \le 2\|E\| \cdot \frac{\pi}{a_0 \times \sigma_{n-r}/2} \quad \text{(by Lemma C.2)}$$
$$= \frac{8\pi\|E\|}{\sigma_{n-r}\lambda_k}. \tag{9}$$

It implies

$$\int_{\Gamma_1} \left\| \sum_{\substack{i \in I_r, j \in I_r^c \\ i \in I_r^c, j \in I_r}} \frac{1}{z(z-\lambda_i)(z-\lambda_j)} u_i u_i^\top E u_j u_j^\top \right\| |dz| \le \frac{8\pi\|E\|}{\sigma_{n-r}\lambda_k}. \tag{10}$$

Combining (5), (8) and (10), we finally obtain

$$M_1 \le \frac{4\pi r^2 x}{\lambda_k^2} + \frac{12\pi\|E\|}{\lambda_k \sigma_{n-r}}. \tag{11}$$

This proves Lemma B.3. For Lemma A.3, we directly have

$$M_1 \le \|E\| \cdot \int_{\Gamma_1} \frac{1}{|z| \times \min_{i \in [n]} |z-\lambda_i|^2} |dz| \le \|E\| \cdot \int_{-T}^{T} \frac{1}{\sqrt{(a_0^2+t^2)} \times (a_0^2+t^2)} dt$$
$$\le \frac{\pi\|E\|}{a_0^2} = \frac{4\pi\|E\|}{\lambda_k^2} \quad \text{(by Lemma C.2)}.$$

By a similar argument, by replacing $\Gamma_1$ by $\Gamma_3 := \{z = a_1 + \mathbf{i}t, -T \le i \le T\}$ and replacing $a_0$ by $a_1$, we have

$$M_3 \le \frac{2\pi r^2 x}{a_1 \delta_{k-k_1}} + \frac{6\pi\|E\|}{a_1 \sigma_{n-r}} = O\left( \frac{r^2 x}{\lambda_{k-k_1}\delta_{k-k_1}} + \frac{\|E\|}{\lambda_{k-k_1}\sigma_{n-r}} \right),$$

and

$$M_3 \le \frac{\pi\|E\|}{a_1 \cdot \delta_{k-k_1}/2} = \frac{4\pi\|E\|}{(\lambda_{k-k_1-1}+\lambda_{k-k_1})\delta_{k-k_1}} \le \frac{4\pi\|E\|}{\lambda_{k-k_1}\delta_{k-k_1}}.$$

This proves Lemma B.4 and Lemma A.4.

## C.2 Estimating integrals over horizontal segments

We are going to bound $M_2$ - integral over the top horizontal segment of $\Gamma_1$ (prove Lemma A.5). The treatment of $M_4$ follows a similar manner. We have

$$N_2 = \int_{\Gamma_2} \frac{1}{|z| \min_{i \in [n]} |z-\lambda_i|^2} |dz|$$
$$= \int_{a_0}^{a_1} \frac{1}{\sqrt{x^2+T^2} \cdot \min_{i \in [n]} ((x-\lambda_i)^2+T^2)} dx \quad \text{(since } \Gamma_2 := \{z \,|\, z = x + \mathbf{i}T, a_0 \le x \le a_1\})$$
$$\le \int_{a_0}^{a_1} \frac{1}{T \cdot T^2} dx$$
$$= \frac{a_1 - a_0}{T^3} \le \frac{1}{T^2}.$$

By similar arguments, we can prove

$$N_4 \le \frac{1}{T^2}.$$

These estimates prove Lemma A.5.

## C.3 Estimating integrals over segment $L$

In this section, we estimate $F_1^{[L]}$, proving Lemma A.6. Recall that

$$F_1^{[L]} := \frac{1}{2\pi} \int_L \left\| z^{-1}(zI - A)^{-1} E(zI - A)^{-1} \right\| |dz|,$$

in which $L := \{t + \mathbf{i}T, b_0 \leq t \leq a_0\}$. Arguing similarly to the previous sections, we also have

$$
\begin{aligned}
\int_L \left\| z^{-1}(zI - A)^{-1}E(zI - A)^{-1} \right\| |dz| &\leq \|E\| \cdot \int_L \frac{1}{|z| \min_{i \in [n]} |z - \lambda_i|^2} |dz| \\
&= \|E\| \cdot \int_{b_0}^{a_0} \frac{1}{\sqrt{t^2 + T^2} \times ((t - \lambda_i)^2 + T^2)} dt \\
&\leq \|E\| \cdot \int_{b_0}^{a_0} \frac{1}{T^3} dt \\
&= \frac{|a_0 - b_0| \|E\|}{T^3}.
\end{aligned}
$$

This proves Lemma A.6.

# D Perturbation bound for inverse low-rank approximations via classical methods

In this section, we present and prove the Eckart–Young–Mirsky-Neumann (EYM–N) bound, as stated in Equation 2 in Section 2. Let $A$ be a symmetric positive definite (PD) matrix with eigenvalues $\lambda_1 \geq \lambda_2 \geq \cdots \geq \lambda_n > 0$, and let $E$ be a symmetric perturbation matrix. Define $\tilde{A} := A + E$, and let $\tilde{\lambda}_1 \geq \tilde{\lambda}_2 \geq \cdots \geq \tilde{\lambda}_n$ be the eigenvalues of $\tilde{A}$. For each $1 \leq p \leq n$, denote by $A_p^{-1}$ and $(\tilde{A}^{-1})_p$ the best rank-$p$ approximations (in spectral norm) of $A^{-1}$ and $\tilde{A}^{-1}$, respectively.

**Theorem D.1 (Eckart–Young-Mirsky–Neumann bound).** *If $4\|E\| \leq \lambda_n$, then*

$$
\|(\tilde{A}^{-1})_p - A_p^{-1}\| \leq \frac{8\|E\|}{3\lambda_n^2} + \frac{2}{\lambda_{n-p}}.
$$

**Proof.** Since $A$ is a PD matrix, $A^{-1}$ is well-defined with the eigenvalues $\lambda_n^{-1} \geq \lambda_{n-1}^{-1} \geq \cdots \geq \lambda_1^{-1} > 0$. Thus, by the Eckart–Young-Mirsky theorem [20], we have

$$
\|A^{-1} - A_p^{-1}\| = \lambda_{n-p}^{-1}.
$$

By Weyl's inequality [62], we have

$$
\|E\| \geq |\lambda_n - \tilde{\lambda}_n| \geq \lambda_n - \tilde{\lambda}_n.
$$

Together with the assumption that $4\|E\| \leq \lambda_n$, this implies

$$
\tilde{\lambda}_n \geq \lambda_n - \|E\| \geq 4\|E\| - \|E\| = 3\|E\| > 0.
$$

Hence, $\tilde{A}$ is also a positive definite matrix. As a result, $\tilde{A}^{-1}$ is well-defined with the eigenvalues $\tilde{\lambda}_n^{-1} \geq \tilde{\lambda}_{n-1}^{-1} \geq \cdots \geq \tilde{\lambda}_n^{-1} > 0$. Thus, similarly, we also have

$$
\|\tilde{A}^{-1} - (\tilde{A}^{-1})_p\| = \tilde{\lambda}_{n-p}^{-1}.
$$

Combining the above equalities with the triangle inequality, we obtain:

$$
\begin{aligned}
\|(\tilde{A}^{-1})_p - A_p^{-1}\| &\leq \|(\tilde{A}^{-1})_p - \tilde{A}^{-1}\| + \|\tilde{A}^{-1} - A^{-1}\| + \|A^{-1} - A_p^{-1}\| \\
&= \tilde{\lambda}_{n-p}^{-1} + \|\tilde{A}^{-1} - A^{-1}\| + \lambda_{n-p}^{-1}.
\end{aligned} \tag{12}
$$

Applying Weyl's inequality again, we further have

$$
\|\tilde{A}^{-1} - A^{-1}\| \geq |\tilde{\lambda}_{n-p}^{-1} - \lambda_{n-p}^{-1}| \geq \tilde{\lambda}_{n-p}^{-1} - \lambda_{n-p}^{-1},
$$

equivalently

$$
\tilde{\lambda}_{n-p}^{-1} \leq \lambda_{n-p}^{-1} + \|\tilde{A}^{-1} - A^{-1}\|. \tag{13}
$$

Together (12) and (13) imply

$$
\|(\tilde{A}^{-1})_p - A_p^{-1}\| \leq 2\left(\lambda_{n-p}^{-1} + \|\tilde{A}^{-1} - A^{-1}\|\right).
$$

To bound $\|\tilde{A}^{-1} - A^{-1}\|$, we use a Neumann series expansion. Under the assumption $\|E\| \le \lambda_n/4$, we have:

$$\|\tilde{A}^{-1} - A^{-1}\| \le \frac{\|E\|}{\lambda_n^2} \cdot \frac{1}{1 - \|E\|/\lambda_n} \le \frac{\|E\|}{\lambda_n^2} \cdot \frac{1}{1 - 1/4} = \frac{4\|E\|}{3\lambda_n^2}.$$

Substituting this bound into the earlier inequality yields:

$$\|(\tilde{A}^{-1})_p - A_p^{-1}\| \le 2\left(\frac{4\|E\|}{3\lambda_n^2} + \frac{1}{\lambda_{n-p}}\right) = \frac{8\|E\|}{3\lambda_n^2} + \frac{2}{\lambda_{n-p}}.$$

∎

# E   Application: Improving the convergence rate of preconditioned conjugate gradient

In this section, we analyze the convergence rate of the preconditioned conjugate gradient (PCG) method for solving $Ax = b$ using an approximately low-rank preconditioner $M$. Without loss of generality, let $A \in \mathbb{R}^{n \times n}$ be symmetric positive definite and $b \in \mathbb{R}^n$.

An effective preconditioner should (i) approximate $A^{-1}$ closely and (ii) accelerate computation. In practice, the exact matrix $A$ is rarely available; instead, one typically works with an approximate version $\tilde{A}$, obtained via rounding or sketching. Although one could in principle set $M = \tilde{A}^{-1}$, computing with a dense inverse is computationally expensive. A more practical choice is therefore

$$M := (\tilde{A}^{-1})_p + \tau U_\perp U_\perp^\top,$$

for some small $\tau > 0$, where $U_\perp U_\perp^\top$ denotes the projection onto the orthogonal complement of the subspace spanned by $(\tilde{A}^{-1})_p$. This *low-rank–plus–regularization* preconditioner has been widely adopted in various schemes, including randomized Nyström preconditioners [22], low-rank correction and deflation methods [8, 10], and low-rank updates for interior-point preconditioners [25].

Let $\hat{x}^{(k)}$ denote the PCG iterate after $k$ steps using $M$, and let $x^* = A^{-1}b$ be the exact solution. A central question is: for a prescribed accuracy $\varepsilon > 0$, how many iterations are required to guarantee

$$\|\hat{x}^{(k)} - x^*\| < \varepsilon\,?$$

Let $E = \tilde{A} - A$, and write $A = \sum_{i=1}^n \lambda_i u_i u_i^\top$, where $\lambda_1 \ge \cdots \ge \lambda_n > 0$ and $\{u_i\}_{i=1}^n$ are orthonormal eigenvectors. For each $1 \le i \le n-1$, define the spectral gap $\delta_i := \lambda_i - \lambda_{i+1}$. Combining the spectral perturbation bound from Theorem 2.1 with the standard PCG residual estimate yields the following result.

**Corollary E.1.** *Under the above setting, for any given $\varepsilon > 0$, if $4\|E\| \le \min\{\lambda_n, \delta_{n-p}\}$, then*

$$\|\hat{x}^{(k)} - A^{-1}b\| < \varepsilon \quad after \quad k = O\left(\sqrt{\frac{\|A\|}{\tau \lambda_n}\left(\frac{\|E\|}{\lambda_n^2} + \frac{\|E\|}{\delta_{n-p}\lambda_{n-p}} + \frac{1}{\|A\|}\right)} \cdot \log(2/\varepsilon)\right) \quad iterations.$$

*Proof.* Let $r_k$ denote the residual at the $k$-th iteration. To achieve $\|\hat{x}^{(k)} - A^{-1}b\| < \varepsilon$, we require

$$\frac{\|r_k\|_A}{\|r_0\|_A} \le \varepsilon, \qquad \text{where } \|x\|_A = \sqrt{x^\top A x}.$$

Using the standard CG residual bound, $\frac{\|r_k\|_A}{\|r_0\|_A} \le 2\left(\frac{\sqrt{\kappa(MA)}-1}{\sqrt{\kappa(MA)}+1}\right)^k$, we require

$$\left(\frac{\sqrt{\kappa(MA)} - 1}{\sqrt{\kappa(MA)} + 1}\right)^k \le \frac{\varepsilon}{2},$$

equivalently,

$$k \ge \frac{\sqrt{\kappa(MA)}}{2} \log(2/\varepsilon). \tag{14}$$

Here, $\kappa(MA) = \|MA\| \cdot \|(MA)^{-1}\|$ is the condition number of $MA$.

*Analyzing $\kappa(MA)$.* Given the decomposition $A = \sum_{i=1}^{n} \lambda_i u_i u_i^\top$, and since $M = (\tilde{A}^{-1})_p + \tau U_\perp U_\perp^\top$, we can write

$$MA = ((\tilde{A}^{-1})_p - (A^{-1})_p)A + \sum_{i=n-p+1}^{n} u_i u_i^\top + \tau U_\perp U_\perp^\top A.$$

Thus,

$$\kappa(MA) = \kappa\left(((\tilde{A}^{-1})_p - (A^{-1})_p)A + \sum_{i=n-p+1}^{n} u_i u_i^\top + \tau U_\perp U_\perp^\top A\right).$$

Since $M$ acts as an approximate inverse on the dominant subspace and scales the complement by $\tau$, the smallest eigenvalue of $MA$ is approximately $\tau \lambda_n$, while the largest is dominated by $1 + \|((\tilde{A}^{-1})_p - (A^{-1})_p)A\|$. Hence,

$$\kappa(MA) \leq O\left(\frac{1 + \|((\tilde{A}^{-1})_p - (A^{-1})_p)A\|}{\tau \lambda_n}\right). \tag{15}$$

In practice, we set $\tau \leq 1/\|A\|$, which ensures the complement contribution $\tau \lambda_1 \leq 1$; with this choice, the bound in (15) holds up to absolute constants. Equality holds approximately when the eigenvectors of $(\tilde{A}^{-1})_p$ align closely with $\{u_{n-p+1}, \ldots, u_n\}$, which is typical since $\tilde{A}$ approximates $A$.

By Theorem 2.1, under $4\|E\| \leq \min\{\lambda_n, \delta_{n-p}\}$,

$$\|(\tilde{A}^{-1})_p - A_p^{-1}\| \leq O\left(\frac{\|E\|}{\lambda_n^2} + \frac{\|E\|}{\delta_{n-p}\lambda_{n-p}}\right).$$

Substituting into (15) yields

$$\kappa(MA) \leq O\left(\frac{\|A\|}{\tau \lambda_n}\left(\frac{\|E\|}{\lambda_n^2} + \frac{\|E\|}{\delta_{n-p}\lambda_{n-p}} + \frac{1}{\|A\|}\right)\right).$$

Substituting this bound into (14) completes the proof. ∎

**Remark E.2.** *By a similar argument, the classical Eckart-Young-Mirsky-Neumann bound* (2) *yields*

$$\|(\tilde{A}^{-1})_p - A_p^{-1}\| \leq \frac{8\|E\|}{3\lambda_n^2} + \frac{2}{\lambda_{n-p}},$$

*leading to the weaker estimate*

$$\kappa(MA) \leq O\left(\frac{\|A\|}{\tau \lambda_n}\left(\frac{\|E\|}{\lambda_n^2} + \frac{1}{\lambda_{n-p}}\right)\right).$$

*Consequently,*

$$\|\hat{x}^{(k)} - A^{-1}b\| < \varepsilon \quad after \quad k = O\left(\sqrt{\frac{\|A\|}{\tau \lambda_n}\left(\frac{\|E\|}{\lambda_n^2} + \frac{1}{\lambda_{n-p}}\right)} \cdot \log(2/\varepsilon)\right) \; iterations.$$

*As discussed in Section 2, Theorem 2.1 improves upon the classical Eckart-Young-Mirsky-Neumann bound by up to a factor of $\sqrt{n}$. Consequently, Corollary E.1 improves the condition number estimate by up to the same factor, corresponding to an improvement of order $n^{1/4}$ in the guaranteed iteration count.*

# F    Maximal allowable variance proxy

Recall from Section 4.1 that, given a matrix $A$, we aim to report the maximal allowable variance proxy $\Delta^{\max}$ for the noise matrix $E$, such that the assumption of Theorem 2.1 holds.

We consider two real-world matrices: the 1990 US Census covariance matrix and the BCSSTK09 stiffness matrix. Each is perturbed by a random matrix $E$ with independent, mean-zero, sub-Gaussian entries and variance proxy $\Delta^2$. It is well known that, with high probability, $\|E\| =$

$(2 + o(1))\Delta\sqrt{n}$, [58, 60]. Thus, for each matrix $A$ and chosen integer $p$, the noise level prescribed by Theorem 2.1 is valid whenever

$$4(2 + o(1))\Delta\sqrt{n} < \min\{\lambda_n, \delta_{n-p}\},$$

which implies the variance proxy must satisfy

$$\Delta^{\max} := \frac{\min\{\lambda_n, \delta_{n-p}\}}{8\sqrt{n}}.$$

Table 1 reports $\Delta^{\max}$ for each $p \in [1, p_A]$, where $p_A$ is the smallest integer such that

$$\frac{\|(A^{-1})_{p_A} - A^{-1}\|}{\|A^{-1}\|} < 0.05.$$

For BCSSTK09 and $p \in \{2, 7\}$, the eigengaps $\delta_{n-p}$ are extremely small (below $10^{-9}$), rendering the corresponding low-rank approximations numerically unstable. We omit $\Delta^{\max}$ in these positions to avoid reporting unreliable values. All numerical values are presented in scientific notation. In each sub-table, we boldface the smallest reported value of $\Delta^{\max}$.

Table 1: Maximal allowable variance proxy $\Delta^{\max} = \min\{\lambda_n, \delta_{n-p}\}/(8\sqrt{n})$ for the 1990 US Census ($n = 69$, $1 \le p \le 17$) and BCSSTK09 ($n = 1083$, $1 \le p \le 8$). Entries are reported in scientific notation. Missing entries correspond to unstable eigengaps.

(a) 1990 US Census ($n = 69$, $1 \le p \le 17$)

| $p$ | $\Delta^{\max}$ | $p$ | $\Delta^{\max}$ |
|---|---|---|---|
| 1 | $\mathbf{2.18 \times 10^1}$ | 10 | $4.78 \times 10^1$ |
| 2 | $4.78 \times 10^1$ | 11 | $4.78 \times 10^1$ |
| 3 | $4.33 \times 10^1$ | 12 | $2.37 \times 10^1$ |
| 4 | $2.75 \times 10^1$ | 13 | $4.43 \times 10^1$ |
| 5 | $4.78 \times 10^1$ | 14 | $3.86 \times 10^1$ |
| 6 | $4.37 \times 10^1$ | 15 | $2.60 \times 10^1$ |
| 7 | $4.78 \times 10^1$ | 16 | $2.79 \times 10^1$ |
| 8 | $4.55 \times 10^1$ | 17 | $4.78 \times 10^1$ |
| 9 | $4.78 \times 10^1$ | | |

(b) BCSSTK09 ($n = 1083$, $1 \le p \le 8$)

| $p$ | $\Delta^{\max}$ |
|---|---|
| 1 | $2.70 \times 10^1$ |
| 2 | $-$ |
| 3 | $2.70 \times 10^1$ |
| 4 | $2.70 \times 10^1$ |
| 5 | $\mathbf{3.83 \times 10^0}$ |
| 6 | $2.70 \times 10^1$ |
| 7 | $-$ |
| 8 | $2.70 \times 10^1$ |

# G  Discretized synthetic Hamiltonian

In this section, we describe the construction of the discretized synthetic Hamiltonian matrix $A$ used in Section 4.2. We begin with the one-dimensional quantum harmonic oscillator:

$$\widehat{H} = -\frac{\hbar^2}{2m}\frac{d^2}{dx^2} + \frac{1}{2}m\omega^2 x^2,$$

whose natural length scale is $\ell := \sqrt{\hbar/(m\omega)}$. Following standard finite-difference benchmarks [55, 38], we truncate the domain to $(-L, L)$ with homogeneous Dirichlet boundary conditions and set $L = 8\ell$.

Let $x_i = -L + i\Delta x$ for $i = 1, \dots, n$, with step size $\Delta x = 2L/(n+1)$, and define the dimensionless grid points $\xi_i = x_i/\ell$ and mesh size $h = \Delta x/\ell$.

A second-order finite-difference discretization of $\widehat{H}$ yields an $n \times n$ real symmetric tridiagonal matrix $H$ with entries

$$H_{ii} = \frac{2}{h^2} + \xi_i^2, \qquad H_{i,i\pm 1} = -\frac{1}{h^2} \quad (1 \le i \le n).$$

**Scaling.** As is standard, we set $\hbar = m = 1$ and $\omega = 4$, so that $\ell = \sqrt{1/4} = 1/2$ [2, 32]. For $h \to 0$, the eigenvalues of $H$ converge to the exact oscillator levels $4i + 2 + \mathcal{O}(h^2)$ for $0 \le i \le n-1$ [55, Program 8], so the discrete spectrum is approximately linear.

Finally, to produce the matrix $A$, whose smallest eigenvalue $\lambda_n$ and gap $\delta_{n-p}$ are compatible with either standard Gaussian or Rademacher noise, we apply the scaling

$$A := 2\sqrt{n}\, H.$$

# H  Empirical sharpness of Theorem 2.1 – numerical results

We numerically report the two scale-free ratios discussed in Section 4.2:

$$\frac{\text{EYM–N bound}}{\text{our bound}} \quad \text{and} \quad \frac{\text{empirical error}}{\text{our bound}},$$

across various configurations $(A, E_k, n, p)$; see Tables 2-9. These ratios reflect the comparative tightness of bounds and are invariant to scaling or normalization. We omit the standard deviations here as they are uniformly small and do not affect interpretation. In all experiments, the ratio $\frac{\text{EYM–N}}{\text{Ours}}$ exceeds 1 across all $C_A$, while $\frac{\text{Empirical}}{\text{Ours}}$ remains consistently around $0.3 - 0.4$.

Table 2: Relative tightness of the EYM–N bound and the empirical error compared to our bound on Census ($n = 69$, $p = 17$) under Gaussian noise.

| $C_A$ | 1.0 | 1.5 | 2.0 | 2.5 | 3.0 | 3.5 | 4.0 | 4.5 | 5.0 | 5.5 | 6.0 |
|---|---|---|---|---|---|---|---|---|---|---|---|
| $\frac{\text{EYM–N}}{\text{Ours}}$ | 5.69 | 3.08 | 3.16 | 2.66 | 2.33 | 2.10 | 1.92 | 1.78 | 1.67 | 1.32 | 1.50 |
| $\frac{\text{Empirical}}{\text{Ours}}$ | 0.433 | 0.395 | 0.454 | 0.473 | 0.401 | 0.461 | 0.443 | 0.427 | 0.435 | 0.402 | 0.446 |

Table 3: Relative tightness of the EYM–N bound and the empirical error compared to our bound on Census ($n = 69$, $p = 17$) under Rademacher noise.

| $C_A$ | 1.0 | 1.5 | 2.0 | 2.5 | 3.0 | 3.5 | 4.0 | 4.5 | 5.0 | 5.5 | 6.0 |
|---|---|---|---|---|---|---|---|---|---|---|---|
| $\frac{\text{EYM–N}}{\text{Ours}}$ | 4.26 | 3.07 | 2.47 | 2.11 | 1.87 | 1.69 | 1.57 | 1.47 | 1.39 | 1.32 | 1.27 |
| $\frac{\text{Empirical}}{\text{Ours}}$ | 0.398 | 0.395 | 0.415 | 0.407 | 0.401 | 0.404 | 0.403 | 0.397 | 0.424 | 0.402 | 0.410 |

Table 4: Relative tightness of the EYM–N bound and the empirical error compared to our bound on BCSSTK09 ($n = 1083$, $p = 8$) under Gaussian noise.

| $C_A$ | 1.2 | 1.4 | 1.6 | 1.8 | 2.0 | 2.2 | 2.4 | 2.6 | 2.8 | 3.0 |
|---|---|---|---|---|---|---|---|---|---|---|
| $\frac{\text{EYM–N}}{\text{Ours}}$ | 3.23 | 2.87 | 2.59 | 2.37 | 2.20 | 2.06 | 1.95 | 1.85 | 1.76 | 1.69 |
| $\frac{\text{Empirical}}{\text{Ours}}$ | 0.320 | 0.325 | 0.353 | 0.372 | 0.334 | 0.347 | 0.318 | 0.348 | 0.306 | 0.354 |

Table 5: Relative tightness of the EYM–N bound and the empirical error compared to our bound on BCSSTK09 ($n = 1083$, $p = 8$) under Rademacher noise.

| $C_A$ | 1.2 | 1.4 | 1.6 | 1.8 | 2.0 | 2.2 | 2.4 | 2.6 | 2.8 | 3.0 |
|---|---|---|---|---|---|---|---|---|---|---|
| $\frac{\text{EYM–N}}{\text{Ours}}$ | 2.48 | 2.22 | 2.03 | 1.88 | 1.75 | 1.66 | 1.57 | 1.50 | 1.44 | 1.39 |
| $\frac{\text{Empirical}}{\text{Ours}}$ | 0.317 | 0.324 | 0.332 | 0.341 | 0.326 | 0.379 | 0.337 | 0.325 | 0.314 | 0.349 |

Table 6: Relative tightness of the EYM–N bound and the empirical error compared to our bound on Discretized Hamiltonian ($n = 500$, $p = 10$) under Gaussian noise.

| $C_A$ | $10^{-4.00}$ | $10^{-3.67}$ | $10^{-3.33}$ | $10^{-3.00}$ | $10^{-2.67}$ | $10^{-2.33}$ | $10^{-2.00}$ | $10^{-1.67}$ | $10^{-1.33}$ | $10^{-1.00}$ |
|---|---|---|---|---|---|---|---|---|---|---|
| $\frac{\text{EYM–N}}{\text{Ours}}$ | 1334 | 618.2 | 287.9 | 133.8 | 62.53 | 29.38 | 13.98 | 6.84 | 3.53 | 1.99 |
| $\frac{\text{Empirical}}{\text{Ours}}$ | 0.301 | 0.326 | 0.331 | 0.346 | 0.321 | 0.318 | 0.339 | 0.338 | 0.309 | 0.371 |

Table 7: Relative tightness of the EYM–N bound and the empirical error compared to our bound on Discretized Hamiltonian ($n = 500$, $p = 10$) under Rademacher noise.

| $C_A$ | $10^{-4.00}$ | $10^{-3.67}$ | $10^{-3.33}$ | $10^{-3.00}$ | $10^{-2.67}$ | $10^{-2.33}$ | $10^{-2.00}$ | $10^{-1.67}$ | $10^{-1.33}$ | $10^{-1.00}$ |
|---|---|---|---|---|---|---|---|---|---|---|
| $\frac{\text{EYM–N}}{\text{Ours}}$ | 946.6 | 439.6 | 204.0 | 95.06 | 44.54 | 21.04 | 10.10 | 5.05 | 2.69 | 1.60 |
| $\frac{\text{Empirical}}{\text{Ours}}$ | 0.315 | 0.327 | 0.326 | 0.310 | 0.331 | 0.317 | 0.308 | 0.340 | 0.307 | 0.309 |

Table 8: Relative tightness of the EYM–N bound and the empirical error compared to our bound on Discretized Hamiltonian ($n = 1000$, $p = 10$) under Gaussian noise.

| $C_A$ | $10^{-4.00}$ | $10^{-3.67}$ | $10^{-3.33}$ | $10^{-3.00}$ | $10^{-2.67}$ | $10^{-2.33}$ | $10^{-2.00}$ | $10^{-1.67}$ | $10^{-1.33}$ | $10^{-1.00}$ |
|---|---|---|---|---|---|---|---|---|---|---|
| $\frac{\text{EYM–N}}{\text{Ours}}$ | 1330 | 617.8 | 287.3 | 133.6 | 62.39 | 29.33 | 13.96 | 6.83 | 3.52 | 1.99 |
| $\frac{\text{Empirical}}{\text{Ours}}$ | 0.317 | 0.334 | 0.373 | 0.336 | 0.347 | 0.310 | 0.322 | 0.314 | 0.330 | 0.349 |

Table 9: Relative tightness of the EYM–N bound and the empirical error compared to our bound on Discretized Hamiltonian ($n = 1000$, $p = 10$) under Rademacher noise.

| $C_A$ | $10^{-4.00}$ | $10^{-3.67}$ | $10^{-3.33}$ | $10^{-3.00}$ | $10^{-2.67}$ | $10^{-2.33}$ | $10^{-2.00}$ | $10^{-1.67}$ | $10^{-1.33}$ | $10^{-1.00}$ |
|---|---|---|---|---|---|---|---|---|---|---|
| $\frac{\text{EYM–N}}{\text{Ours}}$ | 942.0 | 438.0 | 204.0 | 94.80 | 44.30 | 20.90 | 10.10 | 5.03 | 2.69 | 1.60 |
| $\frac{\text{Empirical}}{\text{Ours}}$ | 0.341 | 0.331 | 0.328 | 0.362 | 0.321 | 0.301 | 0.340 | 0.353 | 0.332 | 0.322 |

# I  Examples illustrating limitations of low-rank inverse approximation

This section presents two illustrative examples demonstrating subtle failure modes in low-rank inverse approximation.

## I.1  Eigenvalue reordering due to small eigengaps

We construct an example where a small eigenvalue gap $\delta_{n-p}$ causes the eigenvalues of $\tilde{A}^{-1}$ to reorder, making the low-rank error $\|(\tilde{A}^{-1})_p - A_p^{-1}\|$ a poor proxy for the global error $\|\tilde{A}^{-1} - A^{-1}\|$. In fact, the ratio

$$\frac{\|(\tilde{A}^{-1})_p - A_p^{-1}\|}{\|\tilde{A}^{-1} - A^{-1}\|} \to \infty \quad \text{as } n \to \infty.$$

We illustrate this for $p = 1$ (the construction generalizes). Let $\text{Diag}[a_1, \ldots, a_n]$ denote the diagonal matrix with entries $a_1, \ldots, a_n$. Define:

$$A = \text{Diag}[(4K + 1)\sqrt{n},\ 4K\sqrt{n},\ n, \ldots, n], \quad E = \text{Diag}[\sqrt{n},\ 3\sqrt{n},\ 0, \ldots, 0].$$

Then,

$$\tilde{A} = A + E = \text{Diag}[(4K + 2)\sqrt{n},\ (4K + 3)\sqrt{n},\ n, \ldots, n].$$

The inverses are:

$$A^{-1} = \text{Diag}\left[\frac{1}{(4K + 1)\sqrt{n}},\ \frac{1}{4K\sqrt{n}},\ \frac{1}{n}, \ldots, \frac{1}{n}\right],$$

$$\tilde{A}^{-1} = \text{Diag}\left[\frac{1}{(4K + 2)\sqrt{n}},\ \frac{1}{(4K + 3)\sqrt{n}},\ \frac{1}{n}, \ldots, \frac{1}{n}\right].$$

The best rank-1 approximations retain the top eigenvalue:

$$(\tilde{A}^{-1})_1 = \text{Diag}\left[\frac{1}{(4K + 2)\sqrt{n}},\ 0, \ldots, 0\right], \quad A_1^{-1} = \text{Diag}\left[0,\ \frac{1}{4K\sqrt{n}},\ 0, \ldots, 0\right].$$

Hence,

$$\|(\tilde{A}^{-1})_1 - A_1^{-1}\| = \max\left\{\frac{1}{(4K + 2)\sqrt{n}},\ \frac{1}{4K\sqrt{n}}\right\} = \Theta\left(\frac{1}{4K\sqrt{n}}\right),$$

while

$$\|\tilde{A}^{-1} - A^{-1}\| = \Theta\left(\frac{1}{K^2\sqrt{n}}\right).$$

Choosing $K = n^\varepsilon$, we find the ratio

$$\frac{\|(\tilde{A}^{-1})_1 - A_1^{-1}\|}{\|\tilde{A}^{-1} - A^{-1}\|} = \Theta(n^\varepsilon) \to \infty \quad \text{as } n \to \infty.$$

## I.2 Failure of direct low-rank approximation error to predict inverse error

We now give an example where the low-rank approximation error $\|\tilde{A}_p - A_p\|$ is zero, but the inverse approximation error $\|(\tilde{A}^{-1})_p - A_p^{-1}\|$ grows with $n$.

Let

$$A = \mathrm{Diag}\left[n, \ \frac{1}{2\sqrt{n}}, \ \frac{1}{\sqrt{n}}, \ldots, \frac{1}{\sqrt{n}}\right], \quad E = \mathrm{Diag}\left[0, \ \frac{1}{2\sqrt{n}}, \ \frac{1}{\sqrt{n}}, \ldots, \frac{1}{\sqrt{n}}\right].$$

Then,

$$\tilde{A} = A + E = \mathrm{Diag}\left[n, \ \frac{1}{\sqrt{n}}, \ \frac{2}{\sqrt{n}}, \ldots, \frac{2}{\sqrt{n}}\right].$$

The best rank-1 approximations retain the largest diagonal entry:

$$A_1 = \tilde{A}_1 = \mathrm{Diag}[n, 0, \ldots, 0].$$

Hence, $\|\tilde{A}_1 - A_1\| = 0$. Now consider the inverses:

$$A^{-1} = \mathrm{Diag}\left[\frac{1}{n}, \ 2\sqrt{n}, \ \sqrt{n}, \ldots, \sqrt{n}\right], \quad \tilde{A}^{-1} = \mathrm{Diag}\left[\frac{1}{n}, \ \sqrt{n}, \ \frac{\sqrt{n}}{2}, \ldots, \frac{\sqrt{n}}{2}\right].$$

The rank-1 inverse approximations retain the largest entries:

$$A_1^{-1} = \mathrm{Diag}[0, \ 2\sqrt{n}, \ 0, \ldots, 0], \quad (\tilde{A}^{-1})_1 = \mathrm{Diag}[0, \ \sqrt{n}, \ 0, \ldots, 0].$$

Hence, $\|(\tilde{A}^{-1})_1 - A_1^{-1}\| = \sqrt{n}$. This example shows that even when the direct approximation error $\|\tilde{A}_p - A_p\|$ vanishes, the inverse approximation error can diverge. Consequently, bounding $\|\tilde{A}_p - A_p\|$ alone is insufficient to understand the behavior of low-rank inverse approximations.

## J   Some classical perturbation bounds

This section recalls standard classical results referenced in Section 2, Section 3, and Section D.

**Theorem J.1** (**Eckart-Young-Mirsky bound** [20]). *Let $A, \tilde{A} \in \mathbb{R}^{n \times n}$, and let $A_p$, $\tilde{A}_p$ denote their respective best rank-$p$ approximations. Set $E := \tilde{A} - A$. Then,*

$$\|\tilde{A}_p - A_p\| \leq 2\left(\sigma_{p+1} + \|E\|\right),$$

*where $\sigma_{p+1}$ is the $(p+1)$st singular value of $A$.*

**Theorem J.2** (**Weyl's inequality** [62]). *Let $A, E \in \mathbb{R}^{n \times n}$ be symmetric, and define $\tilde{A} := A + E$. Then, for any $1 \leq i \leq n$,*

$$|\tilde{\lambda}_i - \lambda_i| \leq \|E\| \quad and \quad |\tilde{\sigma}_i - \sigma_i| \leq \|E\|,$$

*where $\lambda_i, \tilde{\lambda}_i$ are the $i$th eigenvalues of $A$ and $\tilde{A}$, and $\sigma_i, \tilde{\sigma}_i$ are the corresponding singular values.*

## K   Notation

This section collects key notations used throughout the paper. Let $A, E$ be symmetric $n \times n$ matrices, and define the perturbed matrix $\tilde{A} := A + E$.

Table 10: Summary of notation used in the paper

| Symbol | Definition |
|---|---|
| $n$ | Dimension of $A$, $\tilde{A}$ |
| $p$ | Target rank parameter |
| $A_p^{-1}$ | Best rank-$p$ approximation to $A^{-1}$ |
| $(\tilde{A}^{-1})_p$ | Best rank-$p$ approximation to $\tilde{A}^{-1}$ |
| $\lambda_1 \geq \cdots \geq \lambda_n$ | Eigenvalues of $A$ in descending order |
| $\tilde{\lambda}_1 \geq \cdots \geq \tilde{\lambda}_n$ | Eigenvalues of $\tilde{A}$ in descending order |
| $\sigma_1 \geq \cdots \geq \sigma_n$ | Singular values of $A$ in descending order |
| $\delta_i$ | $i$-th eigengap: $\delta_i := \lambda_i - \lambda_{i+1}$ |
| $u_i$ | Eigenvector of $A$ corresponding to $\lambda_i$ |
| $\tilde{u}_i$ | Eigenvector of $\tilde{A}$ corresponding to $\tilde{\lambda}_i$ |
| $\Gamma$ | Contour enclosing $\{\lambda_{n-p+1}, \ldots, \lambda_n\}$ (p. 5) |
| $F$ | $\frac{1}{2\pi} \int_\Gamma \|z^{-1}[(zI - \tilde{A})^{-1} - (zI - A)^{-1}]\| \, |dz|$ (p. 5) |
| $F_s$ | $\frac{1}{2\pi} \int_\Gamma \|z^{-1}(zI - A)^{-1}[E(zI - A)^{-1}]^s\| \, |dz|$ (p. 5) |
| $F_1$ | $\frac{1}{2\pi} \int_\Gamma \|z^{-1}(zI - A)^{-1} E(zI - A)^{-1}\| \, |dz|$ (p. 5, Lem. 3.1) |
| $\Delta^{\max}$ | Max. allowable variance: $\frac{\min\{\lambda_n, \delta_{n-p}\}}{8\sqrt{n}}$ (Sec. 4.1) |
| $\mathrm{sr}(A^{-1})$ | Stable rank of $A^{-1}$: $\frac{\|A^{-1}\|_F^2}{\|A^{-1}\|^2}$ (p. 23) |
| Doubling distance $r$ | Smallest $r$ s.t. $2\lambda_{n-p+1} \geq \lambda_{n-r}$ (p. 16, Sec. B) |
| Interaction term $x$ | $\max\limits_{n-r+1 \leq i,j \leq n} |u_i^\top E u_j|$ (p. 16, Sec. B) |
| $\|\cdot\|$ | Spectral norm |
| $\|\cdot\|_F$ | Frobenius norm |
| EYM–N bound | Eckart-Young-Mirsky-Neumann bound |
| PD | Positive semi-definite |

