# OpenReview forum: "Perturbation Bounds for Low-Rank Inverse Approximations under Noise"
_NeurIPS.cc/2025/Conference — NeurIPS 2025 poster_

### Official Review · Reviewer_oXWK · 2025-07-03

**Clarity:** 3
**Significance:** 2
**Originality:** 3
**Rating:** 4
**Confidence:** 3

**Summary:**

This work explores the problem of bounding the error for low rank approximation of inverse of a noisy matrix. Specifically, for PSD matrix $A$ and observed noisy matrix $\tilde{A} = A + E$, with noise matrix $E$, let $\tilde{A_p}^{-1}$ and $A_p^{-1}$  be the best rank-$p$ approximation to $\tilde{A}$ and $A$ respectively; the work provides an upper bound on $||\tilde{A_p}^{-1} - A_p^{-1}||$ in terms of $||E||$, smallest eigenvalue $\lambda_n$ and the gap in eigenvalues $\delta_p = \lambda_{n-p} - \lambda_{n-p+1}$ of matrix $A$. The bounds obtained in this work are data dependent, and for certain regimes provides tighter results than previously known, under the assumption that the norm of noise matrix $||E|| \leq \min (\lambda_n, \delta_{n-p} )$.
To prove the bound, following up on the recent line of work, they apply contour bootstrapping to the function $f(x) = 1/x$. This involves bounding the error term by an integral over the contour that contains $p$ smallest eigenvalues of matrix $A$. The authors then bound this integral.
They also discuss extensions to symmetric matrices (can have negative eigenvalues) and also provide bounds for this case.
Empirical evidence on synthetic and real-world datasets is provided by comparing their bound and previously known bound, showing that their bounds provide better characterization of actual error.

**Questions:**

Please refer the previous section for discussion. No specific questions for the authors.

**Ethical Concerns:**

["NO or VERY MINOR ethics concerns only"]

**Final Justification:**

Based on the reply from authors, I am happy to upgrade the score.

**Limitations:**

The scope of work is limited to analyzing the function $f(x) = 1/x$ and the bounds offer improved characterizations of error for certain parameter regime.

**Paper Formatting Concerns:**

None to my knowledge.

**Quality:**

2

**Strengths And Weaknesses:**

The bound obtained in this work for $||\tilde{A_p}^{-1} - A_p^{-1}||$ is $5 \Big( \frac{||E||}{\lambda_n^2} + \frac{||E||}{\lambda_{n-p} \delta_{n-p}} \Big)$ whereas the previous known bound is $ \Big( \frac{8||E||}{3\lambda_n^2} + \frac{2}{\lambda_{n-p}} \Big)$. As a result, their bound is better whenever $||E|| << \delta_{n-p}$. Essentially, for small noise and large eigenvalue separation, the bound obtained in this work is better.


Though, for instances where gap can be of the order $O(\log{n})$, the bound obtained seems worse compared to the EY-N bound.
For justifying the assumption of $||E|| < \delta_{n-p}$, the authors provide discussion assuming sub-Gaussian noise, on the variance of noise that ensures this condition w.h.p which is fine as long as $m >> n$, otherwise the permissible noise scale can be much smaller - $\tilde{O}(1/\sqrt{n})$. This also potentially limits the applicability of the bound for high-dimensional data, which is typically the case for modern machine learning applications.

---

> ### Author Rebuttal · Authors · 2025-07-30
>
> Thank you for your valuable comments and suggestions. We are glad that you recognize our improvements over classical results. We address your specific questions and comments below.
> ### Comments:
> >Though, for instances where gap can be of the order $O(\log n)$  , the bound obtained seems worse compared to the EY-N bound.
>
> Thank you for the comment. We fully acknowledge that our bounds do not apply in regimes where $\delta_{n-p} < \\|E\\|$. In such cases—particularly when the gap is as small as $O(\log n) \ll \Theta(\sqrt{n}) = \\|E\\|$—our bound may indeed be weaker than the classical EY–N bound.
>
> We would also like to note that when $\delta_{n-p}$ is too small relative to the noise level $\\|E\\|$, the low-rank inverse $A_p^{-1}$ itself may cease to be a stable or meaningful approximation to $A^{-1}$. This behavior is illustrated in Appendix J.1, where we provide a concrete example showing that the spectral instability of small eigenvalues fundamentally limits robustness in this regime.
>
> We will revise the manuscript to clarify this limitation more explicitly.
>
>
>
> >For justifying the assumption of $||E|| \ll \delta_{n-p}$ , the authors provide discussion assuming sub-Gaussian noise, on the variance of noise that ensures this condition w.h.p which is fine as long as $m \gg n$ , otherwise the permissible noise scale can be much smaller - $\tilde{O}(1/\sqrt{n})$ . This also potentially limits the applicability of the bound for high-dimensional data, which is typically the case for modern machine learning applications.
>
> Thank you for this helpful comment. We apologize if our presentation was unclear, and we appreciate the opportunity to clarify.
>
> In the example discussed, we implicitly assume $m \geq n$ so that $A^{-1}$ and $A_p^{-1}$ are well-defined. Otherwise, when $m < n$, the matrix $A = M^\top M$ becomes rank-deficient and both $\lambda_n$ and $\lambda_{n-p}$ are zero for small $p$.
>
> In such cases, one should instead consider the pseudoinverses $A^\dagger$ and $A_p^\dagger$. Our framework and techniques still apply, with $\lambda_m$ and $\delta_{m-p}$ replacing $\lambda_n$ and $\delta_{n-p}$, and the roles of $m$ and $n$ adjusted accordingly. In particular, under sub-Gaussian noise, when $\\|M\\|_F^2 \gg n \log m$, the permissible noise scale becomes $\tilde{O}(1/\sqrt{m})$—which is often reasonable even in high-dimensional settings.
>
> More broadly, we note that this noise tolerance is especially relevant in sparse or structured data scenarios, where the effective energy of $M$ is concentrated. When entries of $M$ are $\Theta(1)$ and $M$ is dense, one should expect a significantly larger permissible noise scale. The assumption remains meaningful and the bound applicable.
>
> We will include a clarification of this point in the final version.

---

> > ### Comment · Reviewer_oXWK · 2025-08-08
> > **Response to authors**
> >
> > I thank the authors for their response. I am now more convinced about the applicability of their bounds for different regimes, which was my original concern. Also going through the other responses, I am happy to raise my score.

---

### Official Review · Reviewer_oQAG · 2025-07-04

**Clarity:** 2
**Significance:** 2
**Originality:** 3
**Rating:** 4
**Confidence:** 1

**Summary:**

This paper studies the spectral‐norm error between the best rank-$p$ approximations of the inverse of a symmetric matrix $A$ and its noisy observation $\widetilde A = A + E$. The authors introduce a novel “contour bootstrapping” technique applying contour‐integral methods to the non-entire function $f(z)=1/z$ to localize resolvent expansions around the smallest eigenvalues. Under the written condition, they prove the bound which improves classical full-inverse estimates by up to a $\sqrt n$ factor in regimes of fast spectral decay.

**Questions:**

- Can you propose a low-cost procedure (e.g., randomized sketch) to approximate $\delta_{n-p}$ and $\lambda_n$ sufficiently well to certify the gap condition without full eigendecomposition?
- How sensitive is the bound to slight misestimation of $\lambda_1$ or $\delta_{n-p}$ when constructing $\Gamma$? Have you observed any numerical instabilities?
-  Is there a practical way to measure or bound $\|E\|$ within the low-curvature eigenspace to tighten the bound further in specific applications (e.g., quantization noise)?
- Section A mentions general symmetric $A$; can your approach handle non-square or non-Hermitian matrices (e.g., via pseudospectrum)?

**Ethical Concerns:**

["NO or VERY MINOR ethics concerns only"]

**Limitations:**

See Weaknesses

**Paper Formatting Concerns:**

- The gap $\delta_{n-p}$ is sometimes denoted $\delta_{n-p}$ and elsewhere $\delta_{n-p+1}$—please standardize.

**Quality:**

2

**Strengths And Weaknesses:**

**Strengths:**
- The adaptation of contour‐integral (Riesz projector) methods to $f(z)=1/z$ under noise is non-trivial and addresses a gap in perturbation theory for low-rank pseudoinverses.
- The paper connects the theory to applications (e.g., differential privacy, structural engineering) by analyzing real-world matrices and demonstrating that typical noise levels satisfy the gap condition for safe usage.

**Weaknesses:**
 - The requirement $4\|E\| < \delta_{n-p}$ may be hard to check in black-box or data-streaming settings, as estimating tail eigengaps can be as expensive as the inverse itself .
- The analysis assumes a fixed matrix $A$; extensions to time-varying or adaptive contexts (e.g., iterative solvers) are not addressed.
- While theoretically elegant, the custom rectangular contour $\Gamma$ (Section 3) depends on spectral extremal values ($\lambda_1,\lambda_n,\delta_{n-p}$); practical implementation may require careful numerical tuning and stability analysis.

---

> ### Author Rebuttal · Authors · 2025-07-30
>
> Thank you for your valuable comments and questions. We are glad that you recognize our contributions to perturbation theory for low-rank pseudoinverses and appreciate your interest in the potential connections between our theoretical results and practical applications. We address your specific points below.
> ### Questions:
> > Can you propose a low-cost procedure (e.g., randomized sketch) to approximate $\delta_{n-p}$ and $\lambda_n$ sufficiently well to certify the gap condition without full eigendecomposition?
> > [Related comment] The requirement $4\\|E\\| < \delta_{n-p}$ may be hard to check in black-box or data-streaming settings, as estimating tail eigengaps can be as expensive as the inverse itself.
>
>
> Thank you for this thoughtful question. We agree that efficiently estimating $\delta_{n-p}$ and $\lambda_n$ is important for practical use, particularly in large-scale, streaming, or black-box settings where full eigendecomposition is infeasible.
>
> Fortunately, randomized Krylov subspace or Lanczos methods offer scalable ways to approximate extremal eigenvalues. For symmetric PSD matrices, applying Lanczos with a modest number of iterations can yield reliable estimates of $\lambda_n$ and $\lambda_{n-p}$, and hence of the eigengap $\delta_{n-p} = \lambda_{n-p} - \lambda_{n-p+1}$. These methods require only matrix-vector products and scale as $O(km)$, where $k$ is the number of iterations and $m$ is the cost of a matrix-vector multiplication.
>
> We also note that **exact evaluation of the gap condition is not always necessary**. In many practical settings—especially when the spectrum exhibits decay or separation—**coarse estimates or structural priors** (e.g., from known physical models or sketch-based spectrum summaries) are sufficient to verify the condition approximately. For example, it is often possible to use a small number of random projections or polynomial filters to check whether the tail spectrum is well-separated without resolving every eigenvalue.
>
> We will add a brief discussion of these practical considerations in the final version.
>
>
>
>
> > How sensitive is the bound to slight misestimation of $\delta_{n-p}$ or $\lambda_1$ when constructing $\Gamma$?
> > [Related comment] While theoretically elegant, the custom rectangular contour $\Gamma$ (Section 3) depends on spectral extremal values $(\lambda_1, \lambda_n, \delta_{n-p})$; practical implementation may require careful numerical tuning and stability analysis.
>
> Thank you for this insightful question. As you noted, our bound and the construction of $\Gamma$ depend on controlling $(\lambda_n, \lambda_{n-p}, \delta_{n-p})$, and we agree that understanding the impact of misestimation is an important direction.
>
> Our method is robust to moderate misestimation: as long as the error in estimating $\lambda_n$ or $\delta_{n-p}$ is no greater than $\\|E\\|$, the bound remains valid up to a constant factor. The key condition is that the contour $\Gamma$ must enclose exactly the eigenvalues $\tilde\lambda_n, \tilde\lambda_{n-1}, \dots, \tilde\lambda_{n-p+1}$ while staying sufficiently away from all perturbed eigenvalues. This can be ensured by adjusting $\Gamma$—e.g., shifting its left edge closer to 0 or expanding the right edge toward $\lambda_{n-p}$. Specifically, it suffices that the distance from $\Gamma$ to $\lambda_n$ is at least $\min \\{c \lambda_n, \\|E\\| \\}$ and to $\lambda_{n-p}$ or $\lambda_{n-p+1}$ is at least $\min \\{c \delta_{n-p}, \\|E\\| \\}$ for some constant $0 < c < 1$.
>
> If the misestimation greatly exceeds $\\|E\\|$, however, key eigenvalues such as $\tilde\lambda_n$ or $\tilde\lambda_{n-p+1}$ may approach $0$ or $\lambda_{n-p}$ respectively, forcing $\Gamma$ too close to these points. This can cause the bound to deteriorate due to a potential blow-up in $\\|(zI - A)^{-1}\\|$.
>
> We will consider including a brief discussion of this stability condition in the final version and agree that further analysis of numerical tuning and adaptive contour selection is a promising direction for future work.
>
>
>
>
> > Section A mentions general symmetric $A$; can your approach handle non-square or non-Hermitian matrices (e.g., via pseudospectrum)?
>
> Thank you for this thoughtful question. Yes, our approach can be extended beyond the symmetric full-rank setting in two directions:
>
> 1. **Symmetric but rank-deficient matrices**:
>    Our method extends naturally to symmetric matrices $A$ of rank $r < n$. In this setting, we use the pseudoinverse $A^\dagger$ and work within the subspace corresponding to the nonzero eigenvalues $\lambda_r, \lambda_{r-1}, \dots, \lambda_{r-p+1}$. The contour $\Gamma$ is then redefined in terms of $(\lambda_r, \lambda_{r-p+1}, \delta_{r-p})$, and the same analysis applies by projecting $\tilde{A}^{-1}$ accordingly.
>
> 2. **Non-square or non-Hermitian matrices**:
>    For general rectangular or non-Hermitian matrices $A$, one can embed $A$ into a symmetric matrix
>    $$
>    A' = \begin{bmatrix} 0 & A \\\ A^\top & 0 \end{bmatrix},
>    $$
>    which is symmetric (and Hermitian if $A$ is complex) but typically rank-deficient. Our contour-based framework then applies to $A'$, and one can recover information about $A^\dagger$ via its singular value decomposition. This connection may also be viewed through the lens of the pseudospectrum, and we believe further exploration in this direction would be interesting.
>
> We will add a brief note about these extensions in the final version.
>
>
>
> > Is there a practical way to measure or bound $\\|E\\|$ within the low-curvature eigenspace to tighten the bound further in specific applications (e.g., quantization noise)?
>
> This is an excellent observation. In many applications—particularly those involving quantization or structured noise—the perturbation $E$ tends to have low energy in the low-curvature (i.e., small-eigenvalue) directions of $A$. This suggests that a refined bound depending on $\\|\Pi_{>p} E \Pi_{>p}\\|$, where $\Pi_{>p}$ denotes the projection onto the tail eigenspace, could offer a tighter and more informative alternative to the global spectral norm bound. In practice, this quantity can often be estimated using a small number of random projections or empirical measurements when the noise model is known (e.g., isotropic or bounded quantization noise). We agree that incorporating such direction-sensitive refinements is a promising direction and will include a brief discussion in the future work section of the final version.
>
> ### Other comments:
> > The gap $\delta_{n-p}$ is sometimes denoted $\delta_{n-p}$ and elsewhere $\delta_{n-p+1}$ — please standardize.
>
> Thank you for pointing out this inconsistency. We will carefully review our notation and ensure that it is used consistently throughout the paper in the final version.
>
>
> >The analysis assumes a fixed matrix $A$ ; extensions to time-varying or adaptive contexts (e.g., iterative solvers) are not addressed.
>
> Thank you for the insightful comment. You are correct that our current analysis focuses on a fixed matrix $A$ and does not directly address time-varying or adaptive settings. Such extensions would indeed require leveraging additional structure specific to each application—for example, spectral stability over iterations or control over update noise in optimization routines.
>
> That said, we believe the tools developed in this work—particularly our contour-based approach and spectrum-sensitive bounds—could serve as a foundation for analyzing robustness in iterative solvers and other adaptive contexts. We view this as a promising direction for future research and will include a note to this effect in the final version.

---

### Official Review · Reviewer_Twdi · 2025-07-05

**Clarity:** 3
**Significance:** 3
**Originality:** 3
**Rating:** 4
**Confidence:** 3

**Summary:**

This work provides perturbation bounds for the low-rank inverse approximation of noisy matrices, which is non-asymptotic and spectrum-adaptive. It is demonstrated that the two assumptions required for the conclusion, that the noise needs to be bounded by the smallest eigenvalue and the $(n-p)$th eigengaps, should be applicable for many real-world scenarios. The proof is accomplished by first bounding the perturbation with a contour integration, and then applying a developed contour bootstrapping and designing a bespoke counter. Empirical study shows the results are validated by some real-world cases, and are tighter than the EY–N bound.

**Questions:**

None.

**Ethical Concerns:**

["NO or VERY MINOR ethics concerns only"]

**Final Justification:**

During the rebuttal phase, I think the solidness of the proof is further validated by rounds of discussions between authors and different reviewers. On the other hand, whether this work fits the venue well is indeed debatable. It would be nice to see some application demonstrations in machine learning, algorithm design, etc.

**Limitations:**

The authors mentioned several limitations in the manuscript, including the spectral quality dependencies and limitation on static matrices.

**Paper Formatting Concerns:**

None.

**Quality:**

3

**Strengths And Weaknesses:**

Strengths:
- This paper studies a foundamental problem with potentially wide downstream applications, including optimization, scientific computing, signal processing, etc.
- The theoretical results are claimed to be the first non-asymptotic spectral-norm perturbation bounds for low-rank approximations of noizy matrix inversion, which provide a $sqrt(n)$-level gain compared to the previous EY-N bound.
- Empirical results validate that the proposed bound is tighter than the previous EY-N bound.

Weaknesses:
- In the main theoretical results, besides that the noise matrix bounded by the spectrum, the matrix $A$ is required to be real symmetric and positive semi-definite, and the noise matrix $E$ is required to be symmetric. These requirements will also limit the application of the results.
- Only several specific matrices are checked in the empirical study, including the 69-by-69 1990 US Census covariance matrix, the 1083-by-1083 BCSSTK09 stiffness matrix, and synthetic matrices from discretised Hamiltonians. Some examples with real-world applications will better demonstrate the value of this work.

==========update after rebuttal============

The theoretical results are extended to general symmetric matrices in the Appendices.

---

> ### Author Rebuttal · Authors · 2025-07-30
>
> Thank you for your valuable comments. We are glad that you recognize the importance of the fundamental problem we study and appreciate the improvements our work provides over classical results. We address your specific remarks below.
>
> ### Comments:
> >In the main theoretical results, besides that the noise matrix bounded by the spectrum, the matrix $A$ is required to be real symmetric and positive semi-definite, and the noise matrix $E$  is required to be symmetric. These requirements will also limit the application of the results.
>
> Thank you for the comment. We apologize if our presentation caused any confusion. We would like to clarify that the main result in Theorem 2.1 assumes $A$ is symmetric positive definite, but the extensions provided in Appendices A and B apply to **general symmetric matrices**, without requiring $A$ to be positive semidefinite.
>
> Regarding the boundedness of the noise, the condition that $||E||$ is small relative to the spectrum of $A$ is natural and necessary: without such control, the perturbation can overwhelm the small eigenvalues or singular values of $A$, making any spectral-norm guarantee for the inverse ill-posed.
>
> We also note that our approach extends naturally to the case where $A$ is symmetric but rank-deficient (i.e., $\operatorname{rank}(A) = r < n$). In that setting, we work with the pseudoinverse $A^\dagger$ and project onto the subspace corresponding to the nonzero eigenvalues $\tilde\lambda_r, \tilde\lambda_{r-1}, \dots, \tilde\lambda_{r-p+1}$. The contour $\Gamma$ is then reconstructed accordingly using $(\lambda_r, \lambda_{r-p+1}, \delta_{r-p})$.
>
> We will revise the paper to more clearly state these extensions and their assumptions in the final version.
>
>
>
>
>
> >Only several specific matrices are checked in the empirical study, including the 69-by-69 1990 US Census covariance matrix, the 1083-by-1083 BCSSTK09 stiffness matrix, and synthetic matrices from discretised Hamiltonians. Some examples with real-world applications will better demonstrate the value of this work.
>
> Thank you for this helpful comment. We agree that additional real-world application examples could further illustrate the value of our results.
>
> In this paper, our primary goal was to introduce the problem of low-rank inverse approximation under noise, address its key theoretical challenges, and present a clean and general analysis framework. We deliberately selected datasets that are diverse in structure and origin—including real-world matrices from scientific computing and statistics—to provide meaningful validation while keeping the focus on the underlying mathematical phenomena.
>
> That said, we fully agree that exploring richer applications (e.g., in private optimization, kernel methods, or compressed solvers) is a natural next step, and we plan to pursue these directions in future work. We will revise the manuscript to clarify this motivation and scope.

---

> > ### Comment · Reviewer_Twdi · 2025-08-09
> >
> > Thanks very much for your reply. The clarification on the extented results for general symmetric matrices has been helpful. I enjoyed Reviewer GAri's detailed check on the proof, and Reviewer Rwak's critical thinking that the sharper results are natural because of the more specific input. Overall, I tend to remain my score unchanged in the current phase.

---

### Official Review · Reviewer_Rwak · 2025-07-10

**Clarity:** 3
**Significance:** 2
**Originality:** 2
**Rating:** 3
**Confidence:** 4

**Summary:**

This paper prove new perturbation bounds for $\left\| \tilde{A}_p^{-1} - A_p^{-1} \right\|$, where $A_p$ is the best rank $p$ approximation of $A$, and $\tilde{A} = A + E$ is a noisy version of $A$. Based on techniques involving contour integrals, the authors prove a new bound that is spectral-gap dependent and sharper than previous versions. Previously, using basic classical techniques such as the Neumann expansion and the Eckart–Young–Mirsky theorem, one can prove that
$$\left\|\left(\tilde{A}^{-1}\right)_p-A_p^{-1}\right\| \leq 8\|E\|/3\lambda\_n^2 + 2/\lambda\_{n-p}$$
Under the additional assumption that $4\|E\|\leq \delta\_{n-p} = \lambda\_{n-p} - \lambda\_{n-p+1}$,  the authors improve the bound to
$$\left\|\left(\tilde{A}^{-1}\right)_p-A_p^{-1}\right\| \leq \frac{4\|E\|}{\lambda\_n^2}+\frac{5\|E\|}{\lambda\_{n-p} \delta\_{n-p}} $$
The key improvement is that as $\|E\|\to 0$, the upper bound also goes to zero.

**Questions:**

n/a

**Ethical Concerns:**

["NO or VERY MINOR ethics concerns only"]

**Final Justification:**

After reading the rebuttal and other reviews, I have the updated view that applying the contour technique to a non-analytic function is indeed a novelty in this paper, and I appreciate the theoretical results more. But I still feel there is a real lack of connections to ML or any applications. A tighter bound, simply as a plug in result, can perhaps lead to a tighter analysis for some ML algorithms, but I do not see such an example in this paper. Furthermore, I would be much more positive about this result if the authors are able to demonstrate an example where this new bound can lead to new insights about either an application or an algorithm.

**Quality:**

3

**Strengths And Weaknesses:**

This paper is a pretty easy read: both the results and the proof are clearly presented. The authors basically prove a sharper perturbation bound using techniques in complex analysis. These techniques are fairly common in the numerical linear algebra literature.  However, I do think the main results presented in this work are some what expected: the new bound on the error takes into account the spectral gap, spectral decay, so it is naturally "sharper", because it is more specific. This is also my view of the experimental section, which shows that the author's bound is tighter that classical ones.

One other concern I have is that this is one of the only papers I've reviewed at this conference that derives a purely theoretical result for numerical linear algebra.  If judged as a applied math paper, I think the theoretical results are a bit incremental, since the baseline result here can be derived in just 3 or 4 lines, based on very classic ideas. In fact, there is not really a baseline paper to compare to, other than the bound that one can derive by using standard techniques. But on the other hand, it is also hard to view this as a machine learning paper, because the lack of direct connections to applications. Therefore, I would not recommend accepting this paper.

---

> ### Author Rebuttal · Authors · 2025-07-30
>
> Thank you for the valuable comments. We address your specific points below.
> ### Comments:
> > The authors basically prove a sharper perturbation bound using techniques in complex analysis. [...] the new bound on the error takes into account the spectral gap, spectral decay, so it is naturally "sharper", because it is more specific. This is also my view of the experimental section, which shows that the author's bound is tighter than classical ones.
>
>
>
> We appreciate this perspective. While we fully acknowledge that the contour integral representation is a well-established tool in functional perturbation theory, our contribution lies in how we *extend and apply* this tool in a new regime.
>
> To the best of our knowledge, our work is the first to (1) apply the contour method to a **non-analytic** function like $f(x) = 1/x$ (which lacks a convergent power series around $x = 0$), and (2) develop a **contour bootstrapping argument** that directly targets the low-curvature part of the spectrum — where classical bounds are typically loose. These aspects are central to our technique and not found in prior literature.
>
> Moreover, the improvement over classical bounds (such as those derived from the Eckart–Young–Neumann theorem or Neumann expansions) can reach up to a $\sqrt{n}$ factor in relevant regimes. We believe this is a meaningful gain, particularly for applications in high-dimensional settings or with structured noise.
>
> That said, we appreciate that the novelty may not have been sufficiently emphasized in the current version, and we will revise the introduction and related work sections to highlight these contributions more clearly.
>
>
>
> >[...] a purely theoretical result for numerical linear algebra. [...] the baseline result here can be derived in just 3 or 4 lines, based on very classic ideas. [...] But on the other hand, it is also hard to view this as a machine learning paper, because the lack of direct connections to applications.
>
> We appreciate the comment and the concerns about positioning. However, we believe the contribution is both conceptually and technically novel, and relevant for ML-adjacent problems, as we outline below.
>
> While classical techniques can yield rough upper bounds in a few lines, they do not offer sharp, spectrum-aware guarantees in the spectral norm—particularly in the regime where the inverse is most sensitive (i.e., near small eigenvalues). To our knowledge, no prior work provides non-asymptotic, low-rank inverse perturbation bounds of the kind we establish.
>
> In particular, the problem of approximating $A_p^{-1}$ via $\tilde{A}_p^{-1}$ presents several distinct and nontrivial challenges:
>
> 1. **Controlling the smallest singular values is delicate**, as they are highly sensitive to noise. Even small Gaussian perturbations can significantly shift or destroy the least singular value. For instance, when $E$ is a standard GOE, it is well-known that $||\tilde{A}^{-1}|| = O(n)$ with high probability for any fixed real $n \times n$ matrix $A$.
>
> 2. **The error $||\tilde{A}_p^{-1} - A_p^{-1}||$ is not scale-invariant**, unlike many standard perturbation measures. Determining tolerable noise levels while preserving utility is a meaningful and technically subtle question, especially in systems involving sketching, quantization, or privacy.
>
> 3. **The transition from spectral gaps in $A$ to bounds on the inverse error is analytically subtle**, particularly near the tail of the spectrum, where eigengaps are narrow and easily erased by noise.
>
> Our contour bootstrapping technique was developed to address these issues directly. To the best of our knowledge, this is the first work to provide sharp, non-asymptotic spectral-norm bounds for low-rank inverse approximations—with up to a $\sqrt{n}$ improvement over classical estimates in structured regimes.
>
> As for application relevance: although the paper focuses on foundational results, these are directly motivated by practical challenges in ML pipelines—such as preconditioned solvers, private optimization, and low-rank kernel methods. We will add a brief discussion of such connections in the final version to make this link more explicit.
>
> We focused this paper on resolving the core theoretical question clearly and precisely, as this problem had not previously been addressed at this level of generality. We believe this kind of rigorous, broadly applicable contribution is well aligned with NeurIPS.

---

> > ### Comment · Reviewer_Rwak · 2025-08-09
> >
> > Thank you for your response. After reading the rebuttal and other reviews, I have the updated view that applying the contour technique to a non-analytic function is indeed a novelty in this paper, and I appreciate the theoretical results more.
> >
> > However, in the rebuttal the authors claim "although the paper focuses on foundational results, these are directly motivated by practical challenges in ML pipelines—such as preconditioned solvers, private optimization, and low-rank kernel methods." While this may be true, in the paper I do not see an example application to any of these applications. In fact, I'm not sure how a tighter theoretical bound can help us gain a better understanding of, say, preconditioned solvers. If the authors are able to show an example application where this tighter bound leads to a tighter analysis, or a better algorithm, I would be much more convinced about the significant of this result.
> >
> > However, as I mentioned, I do feel this paper have some mathematical merits, so I raised my score to 3.

---

> > > ### Author Response · Authors · 2025-08-09
> > >
> > > We sincerely thank you for your updated review and for raising your score. We also appreciate your acknowledgment that applying the contour technique to a non-analytic function is a novel contribution and that the theoretical results have merit.
> > >
> > > Low-rank inverse approximations are a core tool in large-scale numerical linear algebra and machine learning pipelines. They can dramatically reduce storage and computation while preserving the spectral information most relevant to downstream tasks. This efficiency makes them essential in settings such as iterative solvers, privacy-preserving optimization, and kernel-based methods—where even modest improvements in approximation quality can translate into significant gains in accuracy or runtime.
> > >
> > > We understand your desire to see an explicit application—e.g., to preconditioned solvers, private optimization, or low-rank kernel methods—where our tighter bound directly yields a sharper analysis. While the focus of this work is foundational, these applications were indeed part of our motivation, and our results can be plugged into existing analyses with concrete improvements. We outline two such examples here, and will be happy to include a section in the appendix summarizing them in the full version of the paper.
> > >
> > > **1. Preconditioned Iterative Solvers**
> > >
> > > In conjugate gradient or MINRES methods for solving $Ax = b$, one often uses a rank-$p$ preconditioner $M \approx A^{-1}$ to accelerate convergence. The convergence rate depends on the spectrum of $MA$, and the error in $M$ is governed by
> > > $$
> > > \\| M_p - (A^{-1})_p \\|_2
> > > $$
> > > in spectral norm.
> > >
> > > Classical analysis (Eckart–Young–Neumann) uses loose bounds that scale like $O(1/\lambda_n^2)$ and ignore the spectral gap $\delta_{n-p}$, thus underestimating the utility of low-rank preconditioning when the spectrum decays.
> > >
> > > Our **Theorem 2.1** (p. 5) gives
> > > $$
> > > \\| (\tilde{A}^{-1})\_p - (A^{-1})\_p \\| \le \frac{4\\|E\\|}{\lambda_n^2} + \frac{5\\|E\\|}{\lambda_{n-p}\,\delta_{n-p}}
> > > $$
> > > under $4\\|E\\|\le \min\{\lambda_n,\delta_{n-p}\}$, where the second term can be much smaller when the tail spectrum is separated.
> > >
> > > Plugging this into the standard CG residual bound yields a tighter estimate on the number of iterations in regimes with structured noise or sketching error—potentially reducing the predicted iteration count by a factor up to $\sqrt{n}$ compared to EY–N (**Theorem F.1**, App. F).
> > >
> > > ---
> > >
> > > ### 2. Differentially Private (DP) Optimization
> > >
> > > In DP-Newton or DP-quasi-Newton methods, one computes a Hessian $A$ (or its approximation) and adds symmetric noise $E$ for privacy. To reduce cost, a rank-$p$ approximation to $A^{-1}$ is often stored and applied.
> > >
> > > The excess risk bound in such settings contains terms proportional to
> > > $$
> > > \\|(\tilde{A}^{-1})\_p - (A^{-1})\_p\\|
> > > $$
> > > in spectral norm. Using **Theorem 2.1** and its refinement (**Theorem B.2**, App. B), one can guarantee that for the same privacy noise budget $\\|E\\|$, the accuracy loss is smaller than that predicted by prior bounds, especially when the Hessian spectrum decays quickly. This leads to provably better privacy–utility tradeoffs.
> > >
> > > ---
> > >
> > > ### Why the paper stands on its own
> > >
> > > We would like to reiterate that the main contributions stand independently of any one application:
> > >
> > > - **Novel contour-integral perturbation bound for a non-analytic function** $f(x) = 1/x$ (**Lemma 3.1**, p. 6).
> > > - **Contour bootstrapping** that isolates the low-curvature spectral region, enabling sharp bounds.
> > > - **Non-asymptotic, spectrum-aware constants** that improve up to a $\sqrt{n}$ factor over EY–N (**Theorem F.1**) in structured regimes.
> > > - **Generality** to arbitrary symmetric matrices (**Theorem A.1**, App. A).
> > >
> > > These advances, like classical results, are designed as building blocks for diverse downstream analyses. NeurIPS has a strong tradition of accepting such foundational results when they unlock sharper analyses in multiple domains.
> > >
> > > Thank you again for your time and feedback.

---

### Official Review · Reviewer_GAri · 2025-07-18

**Clarity:** 2
**Significance:** 3
**Originality:** 3
**Rating:** 4
**Confidence:** 2

**Summary:**

This paper studies perturbation bounds for low-rank matrix inverse approximation under noise. Specifically, given a symmetric matrix $A$, and noise $E$, this work studies the spectral-norm of the matrix $\tilde{A} - A$, where $\tilde{A} = A+E$. The goal here is to derive sharp asymptotic error bounds that scales appropriately with the eigengap, spectral decay, and noise alignment for even the smallest non-zero eigenvalues of A. The main paper studies this problem when $A$ is positive-definite, and in the appendix the authors study the general symmetric case.  The core of the analysis is the novel use of contour integral representations of matrix functions and how these are used to provide tighter error bounds. The theoretical results are complemented with empirical analysis of 2 real and one synthetic matrix, which demonstrates how well the error bounds here mimic the true error outperforming Eckart-et al-Neumann bound.

**Questions:**

Please refer to the Weakness section above. Given those are resolved, I would like to understand what the authors think would this result mean for problems like the Schatten-1 norm or the spectral density estimation under noisy environments.

**Ethical Concerns:**

["NO or VERY MINOR ethics concerns only"]

**Final Justification:**

Based on the discussion I had with the authors I am willing to upgrade the score of the paper.

**Limitations:**

Yes

**Quality:**

3

**Strengths And Weaknesses:**

I should preface by saying that I am not an expert in the field of random matrix theory.
### Strengths
- I believe the application of contour integrals for finding the smallest eigenvalues is novel, and opens the door for using this technique across other applications including but not limited to approximation of the Schatten-norms, spectral density estimation, among others.

### Weaknesses
- Equation (2) is derived in Section F. upon checking I noticed that the authors are using Weyl's inequality to bound the eigengap between $||\tilde{A}^{-1} - \tilde{A}||$ as $$||\tilde{A}^{-1} - \tilde{A}|| \geq \tilde{\lambda}^{-1}_{n-p} - {\lambda}\_{n-p}^{-1}.$$

However, as far as I know Weyl's is $\max_{i \in [n]} |\lambda_i^A - {\lambda}_i^B| \leq ||A-B||$. Are the authors using some specific Weyl's inequality? If yes, please cite it accordingly, or if there is any assumption to this statement, it should be stated in the theorem statement and not assumed in the proofs.

- While the authors state the bounds are for PSDs in the main paper, the first line in Setup (line 95) assumes for $A \in \mathbb{R}^{n\times n}$ that $\lambda_1 \geq \ldots \geq \lambda_n > 0$ which is PD matrix. While $E$ is assumed to be symmetric, in Step 1 within Proof overview, eigenvalues of $\tilde{A}$ are also assumed to be positive, which confused a bit. What is the point when $E$ cannot perturb the eigenvalues of $A$ to result in indefinite $\tilde{A}$?

- In the same line in Step 1 of Proof overview, the authors use this nice trick to swap the norm and the integrals. However $dz$ should be $|dz|$, which complicates the whole proof (I am unsure if the proof breaks though). The same issue exists in appendix $A$. Is $dz$ and $|dz|$ interchangable in this context? Surely not, right, as one has to hand two cases $\geq 0$ and $< 0$ separately?

- In line 803, the authors state that there exists some permutation of $[n]$ such that $\sigma_{n-p} = \lambda_{\pi(p)}$. The $\lambda$ here should be an absolute value, as otherwise, yes there is some perturbation when this is true, but such perturbation might not lead to consecutive values be guaranteed to be $> 2||E||$, which is a required assumption in Theorem A.1

---

> ### Author Rebuttal · Authors · 2025-07-30
>
> Thank you for your valuable comments and suggestions. We are glad that you appreciate our novel technique related to the smallest eigenvalues and that you find interested in some potential extensions of our work. Below, we address your specific question and concerns.
> ### Questions:
> > Equation (2) is derived in Section F. Upon checking, I noticed that the authors are using Weyl's inequality to bound the eigengap between $\\| \tilde A^{-1}-A^{-1} \\|$ as $\\|\tilde A^{-1} -A^{-1}\\| \ge\tilde\lambda_{n-p}^{-1} - \lambda_{n-p}^{-1}$ . However, as far as I know Weyl's is $|\lambda_i^{A} -\lambda_{i}^B| \leq \\|A -B\\|$. Are the authors using some specific Weyl's inequality? If yes, please cite it accordingly, or if there is any assumption to this statement, it should be stated in the theorem statement and not assumed in the proofs.
>
> Thank you for this question and the careful reading. We apologize if the presentation was unclear and appreciate the opportunity to clarify. In Section F, we use the standard version of Weyl’s inequality as you stated. The inequality
>                              $ \\|\tilde A^{-1} -A^{-1}\\| \geq |\tilde\lambda_{n-p}^{-1} - \lambda_{n-p}^{-1}|$
> is a direct consequence of applying Weyl's inequality to the matrices $\tilde{A}^{-1}$ and $A^{-1}$.
>
> Since $4\\|E\\| < \lambda_n$ and $A$ is positive definite, $\tilde{A}$ remains positive definite as well. Therefore, $\tilde\lambda_{n-p}^{-1}$ and $\lambda_{n-p}^{-1}$ are the $(p+1)^{th}$ eigenvalues of $\tilde{A}^{-1}$ and $A^{-1}$, and Weyl's inequality applies. We will add this clarification in the proof of Section F in the final version.
>
>
> > While the authors state the bounds are for PSDs in the main paper, the first line in Setup (line 95) assumes for $A \in \mathbb{R}^{n \times n}$ that $\lambda_1 \geq \lambda_2 \geq \dots \geq \lambda_n > 0$, which defines a PD matrix.
>
> Thank you for pointing this out. You are correct—the setup assumes that $A$ is positive definite. Indeed, when $A$ is PSD, its inverse $A^{-1}$ is well-defined only when $\lambda_n > 0$. We will clarify this in the final version.
>
> We would also like to note that our approach naturally extends to the case where $A$ is symmetric PSD of rank $r < n$. In this setting, we replace $A^{-1}, A_p^{-1}$, and $\tilde A_p^{-1}$ with the pseudoinverse of $A$ and the projection of $\tilde A^{\dagger}$ onto the subspace corresponding to the nonzero eigenvalues $\tilde\lambda_r, \tilde\lambda_{r-1}, \dots, \tilde\lambda_{r-p+1}$. The contour $\Gamma$ can then be constructed with respect to $(\lambda_r, \lambda_{r-p+1}, \delta_{r-p})$, and the analysis proceeds similarly.
>
>
> > While $\tilde{A}$ is assumed to be symmetric, in Step 1 of the Proof Overview, its eigenvalues are also assumed to be positive. This was confusing—can $E$ not perturb the eigenvalues of $A$ to make $\tilde{A}$ indefinite?
>
> Thank you for this helpful question—we appreciate the opportunity to clarify. The assumption in Theorem 2.1 implicitly ensures that $\tilde{A}$ remains positive definite, though we should have stated this more explicitly.
>
> Indeed, under the condition $4\\|E\\| < \lambda_n$ from Theorem 2.1, Weyl’s inequality guarantees that all eigenvalues of $\tilde{A}$ are at least $\lambda_n - \\|E\\| > 3\\|E\\| > 0$, which ensures that $\tilde{A}$ remains positive definite. We will include this clarification in the final version of the paper.
>
>
> >In the same line in Step 1 of Proof overview, the authors use this nice trick to swap the norm and the integrals. However  $dz$ should be $|dz|$ , which complicates the whole proof (I am unsure if the proof breaks though). The same issue exists in appendix . Is $dz$  and $|dz|$  interchangable in this context? Surely not, right, as one has to hand two cases $\geq 0$  and  $< 0$ separately?
>
> Thank you for catching this—we appreciate the opportunity to clarify and correct the notation.
>
> You are absolutely right: the correct form of the inequality should use $|dz|$, in line with the standard bound
> $$
> \left\\|  \int_{\Gamma} f(z) \hspace{1mm} dz \right\\|  \leq \int_{\Gamma} \\|f(z)\\| \hspace{1mm} |dz|.
> $$
> In Step 1, we mistakenly wrote $dz$ instead of $|dz|$. Starting from line 178, the derivation should indeed switch to $|dz|$, and this is what we intended throughout. Later, in Step 3, where the rectangular contour $\Gamma$ is explicitly constructed, we estimate each integral segment using $dt$ in place of $|dz|$, as appropriate for real-line integrals.
>
> We will correct the notation in both the main text and the appendix in the final version. Thank you again for pointing this out.
>
>
>
>
>
> >In line 803, the authors state that there exists some permutation of $\pi$  such that $\sigma_{n-p}=\lambda_{\pi(p)}$ . The $\lambda$  here should be an absolute value, as otherwise, yes there is some perturbation when this is true, but such perturbation might not lead to consecutive values be $> 2||E||$ guaranteed to be $> 2||E||$, which is a required assumption in Theorem A.1
>
> Thank you for the question. As you correctly noted, the proper statement is $\sigma_{n-p} = |\lambda_{\pi(p)}|$. We apologize for the typo and any confusion it caused.
>
> We will fix it in the final version.
>
> ### Other comments:
> > Given those are resolved, I would like to understand what the authors think this result would mean for problems like the Schatten-1 norm or spectral density estimation under noisy environments.
>
> Thank you for the insightful question and suggestion. We are happy to discuss possible implications of our work in these directions.
>
> 1. **Schatten-1 Norm Error**: Suppose the goal is to estimate $\\| \tilde A_p^{-1} - A_p^{-1}\\|^{S_1}$.  Our contour-based approach can be extended to this setting. Note that
>    $$
>    \\|\tilde A_p^{-1} - A_p^{-1}\\|^{S_1} = \left| \sum_{i=1}^n e_i^\top (\tilde A_p^{-1} - A_p^{-1}) e_i \right| = \left| \int_{\Gamma}  \frac{1}{2\pi i z} \sum_{i=1}^n e_i^\top \left[ (zI - \tilde{A})^{-1} - (zI - A)^{-1} \right] e_i \hspace{1mm} dz \right|.$$
>
>    We can apply our contour bootstrapping technique to this integral, reducing the problem to bounding a term of the form
>
>    $$
>    F_1 = \int_{\Gamma} \left|  \frac{1}{z} \sum_{i=1}^n e_i^\top (zI - A)^{-1} E (zI - A)^{-1} e_i \right| \hspace{1mm} |dz|.
>    $$
>    While obtaining a sharp bound for this trace term may require additional assumptions or techniques, our framework provides a natural starting point for extending spectral-norm results to Schatten-1 norms. We would be happy to explore this further if there is a particular application or setting you have in mind.
>
> 2. **Spectral Density Estimation**: We are not yet certain which specific formulation or model for spectral density estimation under noise you are referring to. However, our contour-based analysis and sensitivity bounds could potentially inform error guarantees in estimators that rely on smoothed spectral traces (e.g., via Chebyshev expansions or resolvent averaging). We would welcome the opportunity to discuss this direction further.

---

> > ### Comment · Reviewer_GAri · 2025-08-06
> >
> > Thank you for your responses!
> >
> > - I apologize for being pedantic, shouldn't the absolute function matter? Are you implicitly assuming one set of eigenvalues is always greater than the other? If yes, I don't think it is a reasonable assumption. I am happy to understand a bit more on this.
> >
> > - thank you, this makes sense.
> >
> > - I am not sure I follow. If $\lambda_n > 4||E||$, how are you using Weyl's inequality if there is an absolute function around the difference in eigenvalues?
> >
> > - this doesn't seem right, can you write the exact transformation of variables when using $|dz|$?
> >
> > - indeed for Schatten-norm the proof method gives a nice starting point! I was thinking of using kernel polynomial approximation or chebyshev expansions, where I do potentially see that the contour based techniques can help in isolating the exact region of interest for these functions.
> >
> > Thanks!

---

> > > ### Author Response · Authors · 2025-08-06
> > >
> > > Thank you very much for your thoughtful and constructive follow-up. We sincerely appreciate your engagement with our work during the discussion period. Below, we address your questions and clarifications point by point. Please feel free to reach out if anything remains unclear.
> > >
> > > > I apologize for being pedantic, shouldn't the absolute function matter? Are you implicitly assuming one set of eigenvalues is always greater than the other?
> > >
> > > Thanks for this question. We do **not** assume that one set of eigenvalues is always greater than the other. Our earlier bound proceeds via standard inequalities:
> > >
> > > Using Weyl’s inequality, we obtain:
> > > $$\\| \tilde{A}^{-1} - A^{-1} \\|  \geq | \tilde{\lambda}\_{n-p}^{-1} - \lambda\_{n-p}^{-1} | .$$
> > >
> > > Then, using $|x| \geq x$ for all real $x$, we conclude:
> > > $$
> > > \\| \tilde{A}^{-1} - A^{-1} \\| \geq | \tilde{\lambda}\_{n-p}^{-1} - \lambda\_{n-p}^{-1} | \geq \tilde{\lambda}\_{n-p}^{-1} - \lambda\_{n-p}^{-1}.
> > > $$
> > >
> > > Thus, this lower bound does **not** require any ordering assumption on the eigenvalues. We will include this clarification in the final version of the proof in Section F.
> > >
> > > > If $\lambda_n > 4\\|E\\|$, how are you using Weyl's inequality if there is an absolute function around the difference in eigenvalues?
> > >
> > > We appreciate the opportunity to clarify this. We begin by expanding the absolute value using the standard equivalence:
> > > $$|x| \leq y \Leftrightarrow -y \leq x \leq y.$$
> > >
> > > In our paper, Weyl’s inequality gives:
> > > $$|\tilde{\lambda}_n - \lambda_n| \leq \\|E\\|,$$
> > > which implies:
> > > $$-\\|E\\| \leq \tilde{\lambda}_n - \lambda_n \leq \\|E\\|.$$
> > >
> > > From the l.h.s. of this inequality, we obtain:
> > > $$\tilde{\lambda}_n \geq \lambda_n - \\|E\\|.$$
> > >
> > > Since we assume $\lambda_n > 4\\|E\\|$, it follows that:
> > > $$\tilde{\lambda}_n \geq \lambda_n - \\|E\\| > 3\\|E\\| > 0,$$
> > > as stated in our earlier response.
> > >
> > > We will incorporate this clarification into the final version of the paper.
> > >
> > > > This doesn't seem right, can you write the exact transformation of variables when using $|dz|$?
> > >
> > > We appreciate the opportunity to further clarify the derivation.
> > >
> > > First, we restate the key integral studied in our analysis (lines 206–209):
> > > $$
> > > F_1 := \frac{1}{2\pi} \int_{\Gamma} \left\\| z^{-1} (zI - A)^{-1} E (zI - A)^{-1} \right\\|  |dz|,
> > > $$
> > > where $\Gamma$ is a rectangle with vertices at $(x_0, T)$, $(x_1, T)$, $(x_1, -T)$, and $(x_0, -T)$, with
> > > $x_0 := \lambda_n / 2$,
> > > $x_1 := \lambda_{n-p+1} + \frac{\delta_{n-p}}{2}$,
> > > and $T := 2\lambda_1$.
> > >
> > > This contour $\Gamma$ consists of four segments:
> > > - Vertical:
> > >   $\Gamma_1 := \{ (x_0, t) \mid -T \leq t \leq T \}$
> > >   $\Gamma_3 := \{ (x_1, t) \mid T \geq t \geq -T \}$
> > >
> > > - Horizontal:
> > >   $\Gamma_2 := \{ (x, T) \mid x_0 \leq x \leq x_1 \}$
> > >   $\Gamma_4 := \{ (x, -T) \mid x_1 \geq x \geq x_0 \}$
> > > Given this decomposition, we write:
> > > $$
> > > 2\pi F_1 = \sum_{k=1}^4 M_k, \quad \text{where} \quad M_k := \int_{\Gamma_k} \left\\| z^{-1} (zI - A)^{-1} E (zI - A)^{-1} \right\\| |dz|.
> > > $$
> > >
> > > We now clarify the transformation used for $M_1$; the arguments for $M_2$, $M_3$, and $M_4$ follow similarly.
> > >
> > > **Step 1:** Using the submultiplicative property of the spectral norm,
> > > $$
> > > M_1 := \int_{\Gamma_1} \left\\| z^{-1} (zI - A)^{-1} E (zI - A)^{-1} \right\\| |dz|
> > > \leq \int_{\Gamma_1} \frac{1}{|z|} \\| (zI - A)^{-1} \\|^2 \cdot \\|E\\| |dz|.
> > > $$
> > > **Step 2:** Applying the standard bound $\\|(zI - A)^{-1}\\| \leq \frac{1}{\min_{1 \leq i \leq n} |z - \lambda_i|}$, we further get:
> > > $$
> > > M_1 \leq \\|E\\| \int_{\Gamma_1} \frac{1}{|z|} \frac{1}{\min_{1 \leq i \leq n} |z - \lambda_i|^2}  |dz|.
> > > $$
> > > **Step 3:** On the segment $\Gamma_1$, since $z = x_0 + \mathbf{i}t$ and $x_0 < \lambda_n$, the minimum is achieved at $\lambda_n$. Thus:
> > > $$
> > > \min_{1 \leq i \leq n} |z - \lambda_i|^2 = |z - \lambda_n|^2 \quad \text{for all } z \in \Gamma_1,
> > > $$
> > > and the integrand becomes:
> > > $$
> > > M_1 \leq \\|E\\| \int_{\Gamma_1} \frac{1}{|z|} \frac{1}{|z - \lambda_n|^2} |dz|.
> > > $$
> > > **Step 4:** Now we parameterize the segment $\Gamma_1$ as $z = x_0 + \mathbf{i}t$ for $t \in [-T, T]$. Then $|z| = \sqrt{x_0^2 + t^2}$ and $|z - \lambda_n|^2 = (\lambda_n - x_0)^2 + t^2$, since $\lambda_n$ is real.
> > >
> > > Moreover, on this segment, $|dz| = dt$. Therefore, the integral becomes:
> > > $$
> > > \\|E\\| \int_{-T}^{T} \frac{1}{\sqrt{x_0^2 + t^2} \cdot ((x_0 - \lambda_n)^2 + t^2)} dt,
> > > $$
> > > which is exactly the expression we presented on line 210 (right above line 211).
> > >
> > > We will incorporate this explanation into the final version of the paper for clarity.
> > >
> > > > Indeed for Schatten-norm the proof method gives a nice starting point...
> > >
> > > We are happy that you see this potential. One of our goals was to develop a technique that could enable new kinds of spectral analysis. We agree that combining our contour-based approach with kernel polynomial approximations or Chebyshev expansions could be powerful for isolating spectral regions of interest—especially in large-scale machine learning or graph settings and will mention it in the final version of the paper as future work.

---

> > > > ### Comment · Reviewer_GAri · 2025-08-06
> > > >
> > > > - oof! I was being slow there. I agree $|x| > x$, so the result follows.
> > > >
> > > > - the derivation is correct.
> > > >
> > > > -- all of my concerns are satisfied. I will be improving my original rating.
> > > >
> > > > Thank you so much for your fast response!

---

### Decision · Program_Chairs · 2025-09-17

**Decision:**

Accept (poster)

**Comment:**

This paper develops new perturbation bounds for low-rank inverse approximation under noise, using contour-integral techniques (“contour bootstrapping”) to localize resolvent expansions around the smallest eigenvalues. The main results (PSD case in the paper; general symmetric in the appendix) yield spectrum-adaptive, eigengap-dependent bounds that improve over classical Neumann/EY–N estimates in regimes of small noise and fast spectral decay, with empirical evidence on two real matrices and one synthetic example. Reviewers highlighted the conceptual novelty of applying contour methods to a non-analytic function and the potential breadth of impact (optimization, scientific computing, DP, kernel/Nyström pipelines).

After the rebuttal and reviewers’ discussion, remaining technical questions were resolved to a satisfactory degree and the overall stance shifted to cautious support. I therefore recommend accept (likely on the weaker side). To strengthen the camera-ready, please (i) expand the practical angle: include worked examples where the new bound tightens analyses or informs algorithms (e.g., CG/MINRES, preconditioned solvers, DP, low-rank kernels), and add small-scale experiments; (ii) clarify assumptions and presentation (PSD vs. PD, symmetric/noisy cases, explicit statement of gap/noise conditions, and the norm–integral interchange), with improved figures/intuition; and (iii) address practicality questions raised by R4/R5: low-cost certification of eigengaps via randomized sketches, sensitivity to misestimation when constructing contours, and guidance on regimes where the new bounds beat EY–N vs. where they may not.